# The large-scale organization of shape processing in the ventral and dorsal pathways

Erez Freud[1,2]*, Jody C Culham[3,4,5], David C Plaut[1,2], Marlene Behrmann[1,2]

[1]Department of Psychology, Carnegie Mellon University, Pittsburgh, United States; [2]Center for the Neural Basis of Cognition, Carnegie Mellon University and the University of Pittsburgh, Pittsburgh, United States; [3]The Brain and Mind Institute, University of Western Ontario, London, Canada; [4]Graduate Program in Neuroscience, University of Western Ontario, London, Canada; [5]Department of Psychology, University of Western Ontario, London, Canada

**Abstract** Although shape perception is considered a function of the ventral visual pathway, evidence suggests that the dorsal pathway also derives shape-based representations. In two psychophysics and neuroimaging experiments, we characterized the response properties, topographical organization and perceptual relevance of these representations. In both pathways, shape sensitivity increased from early visual cortex to extrastriate cortex but then decreased in anterior regions. Moreover, the lateral aspect of the ventral pathway and posterior regions of the dorsal pathway were sensitive to the availability of fundamental shape properties, even for unrecognizable images. This apparent representational similarity between the posterior-dorsal and lateral-ventral regions was corroborated by a multivariate analysis. Finally, as with ventral pathway, the activation profile of posterior dorsal regions was correlated with recognition performance, suggesting a possible contribution to perception. These findings challenge a strict functional dichotomy between the pathways and suggest a more distributed model of shape processing.
DOI: https://doi.org/10.7554/eLife.27576.001

*For correspondence:
erezfreud@gmail.com

## Introduction

Shape is the most fundamental perceived property of objects, and, accordingly, shape processing is crucial for successful visual object recognition (*Palmer, 1999*). Deriving information about the shapes of objects in the input has long been considered to be under the purview of the ventral visual pathway (*Goodale and Milner, 1992*; *Ungerleider and Mishkin, 1982*) and many functional imaging studies have provided evidence to support this claim (e.g., *Freud et al., 2013*; *Grill-Spector et al., 1998*; *Kourtzi and Kanwisher, 2000*; *Lerner et al., 2001*; *Malach et al., 1995*). Specifically, these investigations have uncovered a gradient of shape sensitivity in the ventral pathway, with posterior regions in the early to mid-visual cortex (i.e., V1-hV4) being less responsive to the shape of the object than more anterior regions, such as the Lateral Occipital Cortex (LOC) and the Fusiform Gyrus (FG). Additionally, shape sensitivity in these latter, anterior regions is correlated with perceptual abilities, and a lesion to these areas results in an impairment in object perception (e.g., *Freud et al., 2017a*; *Goodale et al., 1991*; *Konen et al., 2011*).

It appears, however, that shape perception is not solely a product of the computations mediated by the ventral pathway. Human neuroimaging studies (*Freud et al., 2015*; *Jeong and Xu, 2016*; *Konen and Kastner, 2008*; *Zachariou et al., 2014*; *Zachariou et al., 2017*; *Bracci and Op de Beeck, 2016*; *Bracci et al., 2017*), primate electrophysiological studies (*Durand et al., 2007*; *Janssen et al., 2008*; *Janssen et al., 2000*; *Van Dromme et al., 2016*); for review see *Theys et al.,*

**eLife digest** We rely on our sense of vision to perceive the world around us and the objects within it. We also use vision to guide our interactions with objects. One of the most influential theories in cognitive neuroscience is the idea that separate pathways within the brain support these two processes. The ventral pathway is in charge of vision-for-perception. It analyses the features that help us recognize objects, such as their color, size or shape, enabling us to identify the hammer in a toolbox, for example. The dorsal pathway is responsible for vision-for-action. It processes features that help us interact with objects, such as their movement and location, enabling us to use the hammer to strike a nail.

However, recent studies have suggested that the ventral and dorsal pathways may not be as independent as originally thought. Freud et al. now test this idea by examining if the dorsal vision-for-action pathway can also perceive and process objects.

Healthy volunteers viewed pictures of objects while lying inside a brain scanner. Some of the objects in the pictures were intact, whereas others had been distorted. If a brain region shows greater activation when viewing intact objects than distorted ones, it implies that that region is sensitive to the normal shapes of objects. Freud et al. found that both the ventral and dorsal pathways were sensitive to shape, with some areas in the two pathways showing highly similar responses. Furthermore, the shape sensitivity of certain regions within the dorsal pathway correlated with the volunteers' ability to recognize the objects. This suggests that regions distributed across both pathways – and not just the ventral one – may contribute to object recognition.

The two-pathways hypothesis has governed our understanding of vision and of other sensory systems including hearing for several decades. By challenging the binary distinction between the two pathways, the results of Freud et al. suggest that models of sensory processing may require updating. This improved understanding may ultimately improve diagnosis and treatment of perceptual disorders such as agnosia, in which patients struggle to recognize objects.
DOI: https://doi.org/10.7554/eLife.27576.002

2015), comparative studies between human and primates (*Denys et al., 2004*; *Sawamura et al., 2005*) and human neuropsychology (*Freud et al., 2017a*) have all uncovered object representations mediated by the dorsal pathway, even under conditions in which no visuomotor response is required. However, many questions remain concerning the topographical organization of the dorsal object representations, the nature of the visual properties encoded in these representations and their contribution to perceptual behavior.

In the current study, we addressed these questions in two studies, each combining functional magnetic resonance imaging (fMRI) and psychophysical measures. We hypothesized that the large-scale topographical organization of dorsal object representations obeys a spatial gradient in which posterior regions of the pathway derive shape representations and contribute to visual perception, whereas more anterior regions derive representations that are better tuned to subserve visuomotor behaviors (*Freud et al., 2016*). Our motivation stems from two key issues concerning dorsal pathway function: the first is the need to reconcile recent findings of dorsal pathway activation under conditions of perception (as noted above) with the well-established findings that the dorsal pathway is engaged in object-directed actions such as grasping and manipulation (e.g., *Culham et al., 2003*; *Fabbri et al., 2016*; for a review, see *Gallivan and Culham, 2015*). The second concerns the connectivity that anchors the parietal representational continuum at one end with more caudal regions connected to visual cortex (*Greenberg et al., 2012*) and, at the other end, with more rostral regions anatomically and functionally connected to motor cortex (*Davare et al., 2010*).

We adopt an approach that has been used previously to characterize the neural basis of shape processing in ventral cortex in response to stimuli in which shape information is parametrically eliminated by increasingly distorting images of objects (*Lerner et al., 2001*) (see *Figure 1A* for example). The decrease in BOLD activation with increased scrambling serves as an index of shape sensitivity (*Denys et al., 2004*; *Grill-Spector et al., 1998*; *Lerner et al., 2001*; *Malach et al., 1995*; *Murray et al., 2002*).

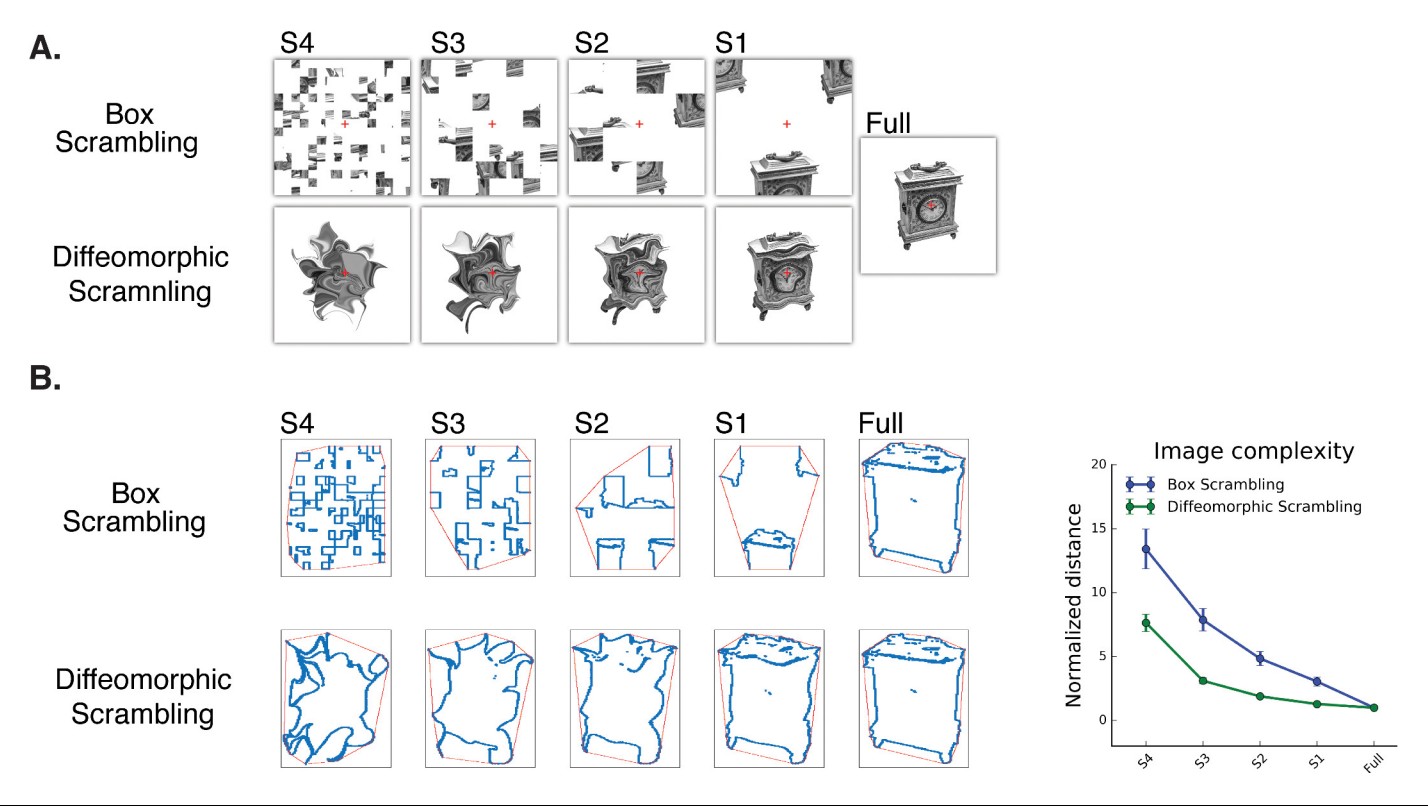

**Figure 1.** Experimental stimuli (**A**) and a quantitative analysis of 'goodness of shape' following the different scrambling procedures (**B**). (**A**) Box scrambling experiment (upper row). Shape information was altered by dividing the display using an invisible grid and then randomly rearranging the squares. The diffeomorphic scrambling experiment (bottom row) distorted the outer contour and distinctive features of the object, while preserving the existence of a single shape but rendering it increasingly unrecognizable. This is achieved by repeatedly applying a flow field generated from a set of two-dimensional cosine components with random phase and amplitude (*Stojanoski and Cusack, 2014*). Both methods of scrambling preserve low-level factors of the image (e.g., average luminance, number of pixels). (**B**) Image analysis as a function of scrambling method and scrambling level. For each image, the minimal distance between the shape edges (blue dots) and the image's convex hull (red frame) was computed and normalized relative to the full (intact) image. Although both manipulations resulted in reduced shape information, a greater decrease in the availability of shape information was found in the box scrambling experiment compared to the diffeomorphic scrambling experiment. Similar results were obtained when other image analysis algorithms were employed, suggesting that the amount of shape information (including texture, figure-ground segregation and defined edges) was greater for the diffeomorphic than scrambled stimuli (see methods for details).

DOI: https://doi.org/10.7554/eLife.27576.003

Following these previous studies, in the first experiment, we adopted a parametric box scrambling procedure to map the large-scale organization of shape processing along the ventral and dorsal visual pathways and the contribution of these representations to shape perception. In addition, we employed more advanced multivariate analyses to further elucidate the nature of representations and the similarities or differences in these representations in the two pathways. However, the box-scrambling method has some inherent limitations. First, it increases the number of edges, and early visual cortex is especially sensitive to this type of information (*Grill-Spector et al., 1998*; *Lerner et al., 2001*). Second, box scrambling disrupts crucial shape attributes such as good continuation and figure-ground segregation (*Koffka, 1935*; *Qiu and von der Heydt, 2005*; *Read et al., 1997*), and, unsurprisingly, eliminates the identity information of the stimulus. Furthermore, in the box scrambling experiment, participants were shown each object at every level of scrambling and so there may have been priming or adaptation for the same object across levels of scrambling.

To circumvent these limitations and verify the results, in the second experiment we used a diffeomorphic scrambling manipulation that increasingly precludes the identification of the object but preserves some fundamental shape properties by repeatedly applying a flow field generated from a set of two-dimensional cosine components with random phase and amplitude (*Stojanoski and Cusack,*

*2014*). Consequently, some shape information (i.e., the presence of an object or figure that is clearly differentiated from the background and has well-defined boundaries, relatively uniform texture and center of mass; see methods and *Figure 1B* for quantitative analyses of object distortion) is still largely available even in the most distorted version, but the object itself is parametrically distorted and unrecognizable. This manipulation enabled us to re-examine the large-scale organization of the two pathways, and to disentangle sensitivity to shape versus identity information. Last, each object was presented at only one level of distortion (counterbalanced across participants) to remove possible priming effects.

## Results

The large-scale organization of shape processing along the dorsal and ventral visual pathways was assessed in two imaging experiments, by computing the level of the BOLD signal as a function of the decrease in shape information across five levels. Three analytical approaches were employed: a novel voxel-wise approach that provides a continuous voxel-wise mapping of shape processing along the two pathways, a more traditional ROI analysis, and a multivariate representational similarity analysis (RSA) to uncover additional information about the similarity of spatial patterns of activation as a function of scrambling.

### Univariate analysis

#### Box scrambling experiment

To map the topographical organization of shape processing along the two pathways in a continuous fashion, we generated a voxel-wise map of shape sensitivity for all visually selective voxels. In each voxel, beta weights were extracted for each of the five stimulus conditions from most scrambled to intact (S4, S3, S2, S1, Full), and the linear slope between these conditions was used as an index of shape sensitivity. Voxels that are shape sensitive were those in which activation, as a function of object coherence, had a positive slope. Conversely, voxels that showed greater sensitivity for scrambled than intact images had a negative slope. To map the profile of shape sensitivity across all voxels, we then applied a piecewise linear regression across all the voxels using two (rather than one) linear components (see Methods for details). In each pathway, the piecewise linear regression increased the $R^2$ when compared with a simple linear regression (ps <0.05), thereby providing a better fit to the data than a one-component linear model (see Materials and methods).

In the ventral pathway (*Figure 2A*), a negative slope was found in the vicinity of the calcarine sulcus and in the posterior occipital lobe more generally, and likely reflects the sensitivity of early visual cortex to edges and high spatial frequency information in the input (*Lerner et al., 2001*); for an additional interpretation, see *Murray et al., 2002*). In more anterior regions of the ventral pathway, a positive slope was evident both on the lateral and the inferior surface of the occipitotemporal cortex (i.e., LO, Parahippocampal gyrus and Fusiform gyrus). However, in even more anterior regions (i.e., anterior and medial temporal cortex), a decrease in shape sensitivity was detected. The piecewise regression revealed that the data were well described by two linear components. The first component reflected the increase in shape sensitivity in the transition between early visual cortex and object-selective cortex and was characterized by a positive correlation between slope and location on the posterior-anterior axis [LH: $t_{(10)}$=8.21, q < 0.00001 CI {0.39, 0.69}; RH: $t_{(10)}$=12.1 q < 0.00001, CI {0.43, 0.62}]. The second component was characterized by a decrease in shape sensitivity, as evident from the negative correlation between location on the posterior-anterior axis and slope [LH: $t_{(10)}$=9.5, q = 0.00001, CI{−0.32, −0.51}; RH: $t_{(10)}$=6.94, q = 0.00001, CI {−0.20 −0.39}; *Figure 2B*]. The robust positive first component is consistent with previous investigations of the ventral pathway (*Lerner et al., 2001*; *Murray et al., 2002*), and replicates the hierarchical nature of shape processing in which more elaborate and complex representations are derived as one moves rostrally (*Grill-Spector and Weiner, 2014*). The reduction in shape sensitivity in the anterior parts of the temporal lobe might reflect a transition to more semantic and memory-based representation (*Kravitz et al., 2013*; *Visser et al., 2010*), rather than shape sensitivity per se, but might also be a consequence of MRI distortion effects which are more common in these regions (*Olman et al., 2009*).

As in the ventral pathway, shape sensitivity in the dorsal pathway was not evident in early visual areas, but emerged in extrastriate occipital cortex and reached a peak in the posterior intraparietal sulcus (IPS). Consistent with the findings of the ventral pathway, shape selectivity gradually

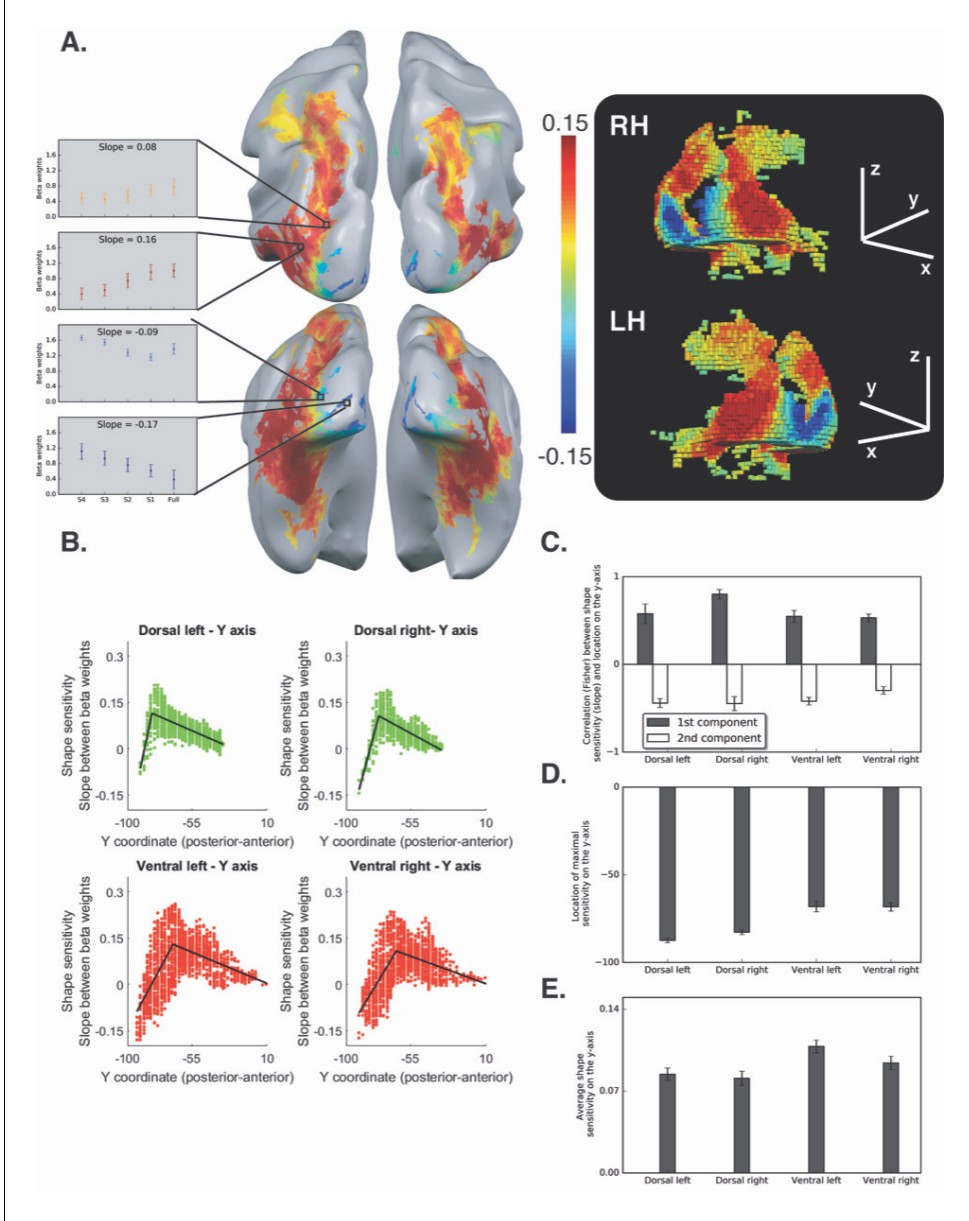

**Figure 2.** Voxel-wise analysis for the box scrambling experiment. (A) Shape sensitivity is projected on an inflated brain from a superior view (upper panel) and from a posterior-inferior view (lower panel). Warm colors signify voxels that are shape sensitive, with activation increasing as a function of object coherence. Conversely, cold colors reflect low shape sensitivity (negative slopes) or greater sensitivity for scrambled than intact images. The activation profile of four representative clusters (10 voxels each) is plotted and the color of the bars reflects the slope value of each cluster (left panel). The right panel is a 3D reconstruction of the shape sensitivity of all visual voxels along the two pathways. (B) Group-averaged piecewise regression analysis. Each dot signifies a voxel, at a particular y-coordinate, averaged across participants, and the black line shows the result of the piecewise regression (based on two linear components) obtained for the group average. In both the dorsal and ventral pathways, the first component showed that the voxel location on the posterior-anterior axis was significantly correlated with shape sensitivity, which depicts how shape selectivity increases moving forward from the occipital pole through extrastriate and inferior temporal areas. The second component reflects a decrease in shape selectivity. (C) Correlation coefficients for each component, computed for individual participants, reveal that the large-scale organization of the two pathways was reliable across participants. (D) The point of maximal-shape sensitivity (inflection point) differed between the two pathways. This was more posterior in the dorsal versus ventral pathway. (E) The average shape sensitivity of all shape-selective voxels was greater in the ventral than dorsal pathway.
*Figure 2—figure supplement 1* (upper panel) shows that similar results were observed when the analysis excluded pictures of tools and was restricted to images of objects with no visuomotor association. *Figure 2—figure supplement 1* (lower panel) shows that similar results were obtained when the piecewise regression was based on distance that was calculated from the combination of the Y and Z coordinates (see *Figure 2C* for a comparison).
*Figure 2—figure supplement 2* shows the ROI analysis in which slope (shape sensitivity) is plotted as function of Region of Interest defined from atlases, separately for each pathway and hemisphere. Black asterisks signify that a ROI is significantly sensitive to shape (slope >0, q < 0.05 FDR

*Figure 2 continued on next page*

*Figure 2 continued*

corrected). The black vertical line separates the lateral and inferior ROIs of the ventral pathway. Error bars in all graphs represent the standard errors. For all figures, see also source data files for individual data points.

DOI: https://doi.org/10.7554/eLife.27576.004

The following source data and figure supplements are available for figure 2:

**Source data 1.** Individual data points for *Figure 2C*.
DOI: https://doi.org/10.7554/eLife.27576.007
**Source data 2.** Individual data points for *Figure 2D*.
DOI: https://doi.org/10.7554/eLife.27576.008
**Source data 3.** Individual data points for *Figure 2E*.
DOI: https://doi.org/10.7554/eLife.27576.009
**Source data 4.** Individual data points for *Figure 2—figure supplement 1* (upper panel).
DOI: https://doi.org/10.7554/eLife.27576.010
**Source data 5.** Individual data points for *Figure 2—figure supplement 1* (lower panel).
DOI: https://doi.org/10.7554/eLife.27576.011
**Source data 6.** Individual data points for *Figure 2—figure supplement 2*.
DOI: https://doi.org/10.7554/eLife.27576.012
**Figure supplement 1.** Two components analysis for the box scrambling experiment.
DOI: https://doi.org/10.7554/eLife.27576.005
**Figure supplement 2.** Shape sensitivity for Dorsal left and Dorsal right and Ventral left and Ventral right.
DOI: https://doi.org/10.7554/eLife.27576.006

decreased from the posterior parietal lobe to more anterior regions (*Figure 2A*). Accordingly, the piecewise regression revealed a positive correlation for the first component, which reflects the increase in shape sensitivity [LH: $t_{(10)}$=5.15, q = 0.0004 CI {0.32, 0.82}; RH: $t_{(10)}$=10, q = 0.00001, CI {0.62, 0.97}], and a robust negative correlation for the second component, which reflects the decrease in shape sensitivity in the anterior parts of the dorsal pathway [LH: $t_{(10)}$=8.54, q = 0.00001 CI {−0.32, −0.55}; RH: $t_{(10)}$=8.94, q = 0.000001, CI {−0.34, −0.57}; *Figure 2C*].

The key finding here is that the two pathways have similar topographical organization. In addition, these findings provide novel evidence for the nature of shape processing in the dorsal pathway and indicate that object representations in this pathway are not monolithic but, rather, differ qualitatively along the posterior-anterior axis (*Freud et al., 2016*).

The mapping of topographical organization of shape sensitivity was done along the posterior-anterior axis. However, given the curvature of the two pathways in the brain, it might be the case that shape information is also modulated by the inferior-superior axis. To examine this possibility, we calculated the distance between each voxel to the most posterior voxel using both Y and Z coordinates and then evaluated the shape-sensitivity gradient. This analysis replicates the organization revealed using only the Y-axis and indicates the Y-axis serves as the key dimension (see *Figure 2—figure supplement 1*, lower panel).

Despite the apparent qualitative similarity in the large-scale topographical organization of shape processing along the two pathways, further analyses also revealed some differences between the ventral and dorsal pathways. In particular, the maximal shape sensitivity (the apex or inflection point of the regression line) was more posterior in the dorsal relative to the ventral pathway [$F_{(1,10)}$ = 22.11, p=0.0008, $\eta_p^2$ = 0.68, CI {12 21}; *Figure 2D*]. This effect was reproduced when the main dependent variable was the distance from the most posterior voxel based on the Y and Z coordinates [$F_{(1,10)}$ = 91, p=0.000002, $\eta_p^2$ = 0.90, CI {16 27}].

Additionally, the average sensitivity of all shape-selective voxels (slope > 0) was significantly greater in the ventral than dorsal pathway [$F_{(1,10)}$ = 45, p=0.00005, $\eta_p^2$ = 0.81, CI {0.01 0.02}; *Figure 2E*]. Interestingly, this main effect was modulated by an interaction with hemisphere [$F_{(1,10)}$ = 11.37, p=0.007, $\eta_p^2$ = 0.53, CI {0.01 0.02}] with greater difference between the pathways in the left than the right hemisphere. The greater difference between the pathways in shape sensitivity in the left hemisphere might reflect hemispheric specialization, such that the left dorsal pathway is more tuned to visuomotor aspects than the right dorsal pathway.

Finally, in addition to the continuous analysis described above, the data were analyzed using an ROI approach. ROIs were defined based on a probabilistic atlas (*Wang et al., 2015*) and the aIPS

ROI was defined based on a meta-analysis (Neurosynth.org). This analysis reproduced the results reported above and demonstrated the two components organization of the two pathways (*Figure 2—figure supplement 2*).

## Diffeomorphic scrambling experiment

The second experiment utilized the diffeomorphic stimuli to replicate the large-scale organization observed for the box scrambling method as well as to provide further insight into the nature of representations derived by the different regions.

In the ventral pathway, the piecewise regression revealed a positive correlation between slope and location on the posterior-anterior axis (first component), which reflects the emergence of shape sensitivity in the transition from early visual cortex to object selective cortex [LH: $t_{(10)}$=25.2 q<0.0000001, CI [0.59, 0.70]; RH: $t_{(10)}$=13.2, q = 0.000004, CI {44, 0.63}], and a negative correlation for the second component, which reflects the decrease in shape sensitivity in the most anterior parts of the temporal lobe [LH: $t_{(10)}$= 9.77, q = 0.0000005, CI{−0.25, −0.40}; RH: $t_{(10)}$=4.57, q = 0.001, CI {−0.09, −0.27}].

These findings replicate those of the box scrambling experiment, in which sensitivity to shape information was not present in posterior regions of the ventral pathway, increased in more rostral parts of the pathway and then decreased in the more anterior temporal regions. Interestingly, however, on the lateral surface of the ventral pathway (corresponding to LOC, see also ROI analysis, *Figure 3—figure supplement 2*), a flatter slope was found (i.e., similar activation to the different levels of image distortion), suggesting that this region was less sensitive to the diffeomorphic manipulation than to the box scrambling manipulation. Consistent with previous reports (*Malach et al., 1995*; *Margalit et al., 2016*), LOC appears to represent the presence of a well-defined shape, with clear contours, rather than a representation of the object's identity (*Figure 3A*).

The piecewise regression of the dorsal pathway revealed two opposite linear components. The first component was characterized by a positive correlation between slope and location on the y-axis, which reflects the transition from early visual cortex to object-selective cortex [LH: $t_{(10)}$=3.98, q = 0.002, CI {0.22, 0.80}; RH: $t_{(10)}$=7.31, q = 0.00005, CI{43, 0.81}]. In contrast, the second component, which reflects the representational gradient along the IPS, was characterized by a negative correlation between slope and location of the voxels on the posterior-anterior axis [LH: $t_{(10)}$=5.96, q = 0.0001, CI {−0.16, −0.35}; RH: $t_{(10)}$=6.39, q = 0.0001, CI{−0.19, −0.39}].

A repeated measures ANOVA on the correlation values (between slope and location of voxels on the y-axis), with Experiment as a between-subjects variable, and pathway, component and hemisphere as within-subject variables, revealed no interactions between Experiment and the other variables [all ps > 0.05]. This result indicates that, despite the difference in the overall shape sensitivity between the two experiments, they provided consistent information on the overall large-scale organization of shape sensitivity.

As in box scrambling experiment, the maximal shape sensitivity emerged in more posterior regions of the dorsal pathway compared with the ventral pathway [$F_{(1,10)}$=18.78, p=0.001, $\eta_p^2$ = 0.65], and the effect held when distance based on the Y and Z axes served as the dependent variable [$F_{(1,10)}$=45, p=0.00005, $\eta_p^2$ = 0.81]. The similarity between the two experiments was confirmed by an ANOVA that included Experiment as a between-subjects variable. In particular, a main effect of pathway was found [$F_{(1,20)}$=56, p=0.000001, $\eta_p^2$ = 0.73] and was not modulated by experiment [$F_{(1,20)}$ < 1]. Interestingly, the inflection point of the regression was more anterior in the diffeomorphic scrambling experiment compared with box scrambling experiment [$F_{(1,20)}$=8.45, p=0.008, $\eta_p^2$ = 0.29], and this might reflect the increased high-level shape or identity processing associated with the diffeomorphic scrambling stimuli.

In addition, we compared the overall neural sensitivity to shape information in all shape-selective voxels (slope > 0). The average shape sensitivity was greater in the ventral than the dorsal pathway [$F_{(1,10)}$=37, p=0.0001, $\eta_p^2$ = 0.79]. The comparison between the two experiments revealed two main effects, with no interaction. As expected, there was a main effect of pathway: shape sensitivity was greater overall in the ventral than dorsal pathway across the two experiments [$F_{(1,20)}$=83, p=0.000001, $\eta_p^2$ = 0.8]. There was also an additional main effect of Experiment, with greater average positive slope in the box scrambling experiment than the diffeomorphic scrambling experiment, reflecting the different nature of the two manipulations [$F_{(1,20)}$=32, p=0.00001, $\eta_p^2$ = 0.62; see also

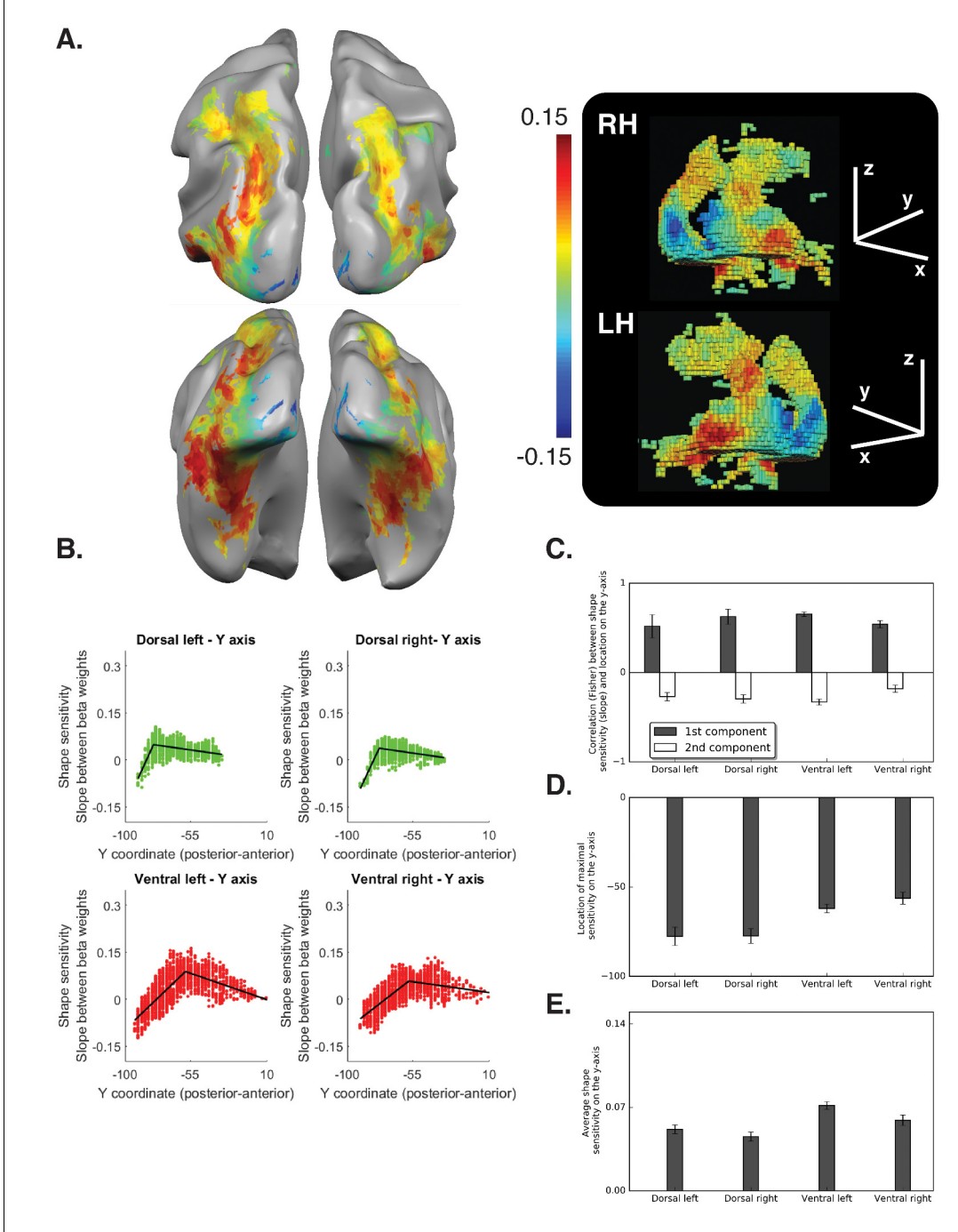

**Figure 3.** Diffeomorphic scrambling experiment Voxel-wise analysis. (**A**) Warm colors signify voxels that are shape sensitive, that is, voxels in which activation, as a function of object coherence, had a positive slope. Conversely, cold colors reflect low shape sensitivity (negative slopes) or greater sensitivity for scrambled than intact images. The right panel is a 3D reconstruction of all visual voxels along the two pathways. (**B**) Group-averaged piecewise regression analysis. Each dot signifies the mean of a single voxel, averaged across participants, and the black line shows the result of the two-components piecewise regression obtained for the group average. In agreement with the box scrambling experiment, the location of a voxel on the posterior-anterior y-axis in both the dorsal and ventral pathways was positively correlated with slope, and the second component was characterized by a robust negative correlation, suggesting that shape sensitivity decreased in more rostral parts of the two pathways. (**C**) Correlation coefficients for each component computed for individual participants reveal that the large-scale organization of the two pathways was reliable across participants. (**D**) The point of maximal-shape sensitivity (inflection point) differed between the two pathways and was more posterior in the dorsal versus ventral pathway. (**E**) The average shape sensitivity of all shape-selective voxels was greater in the ventral pathway compared with the dorsal pathway.
*Figure 3—figure supplement 1* (upper panel) shows that similar results were observed when the analysis excluded pictures of tools and was
*Figure 3 continued on next page*

*Figure 3 continued*

conducted only on objects with no visuomotor association. *Figure 3—figure supplement 1* (lower panel) shows that similar results were obtained when the piecewise regression was based on distance that was calculated from the combination of the Y and Z coordinates. *Figure 3—figure supplement 2* (lower panel) aims to compare between the ROIs analysis of the two experiments and shows the ROI analysis in which slope (shape sensitivity) is plotted as function of Region of Interest defined from atlases, separately for each pathway and hemisphere. Black and gray asterisks signify that a ROI is significantly sensitive to shape (slope >0, q < 0.05) in the box scrambling experiment and the diffeomorphic scrambling experiment, correspondingly. Gray-filled circles (q < 0.05) and gray-filled triangles (q < 0.1) signify that, in a particular ROI, shape sensitivity was greater in the box scrambling experiment compared with the diffeomorphic experiment. The black vertical line separates the lateral and inferior ROIs of the ventral pathway. Along the dorsal pathway, most ROIs were more shape sensitive in the box scrambling experiment compared with the diffeomorphic experiment. In contrast, in the ventral pathway only the lateral ROIs showed this distinction.

DOI: https://doi.org/10.7554/eLife.27576.013

The following source data and figure supplements are available for figure 3:

**Source data 1.** Individual data points for *Figure 3C*.
DOI: https://doi.org/10.7554/eLife.27576.016
**Source data 2.** Individual data points for *Figure 3D*.
DOI: https://doi.org/10.7554/eLife.27576.017
**Source data 3.** Individual data points for *Figure 3E*.
DOI: https://doi.org/10.7554/eLife.27576.018
**Source data 4.** Individual data points for *Figure 3—figure supplement 1* (upper panel).
DOI: https://doi.org/10.7554/eLife.27576.019
**Source data 5.** Individual data points for *Figure 3—figure supplement 1* (lower panel).
DOI: https://doi.org/10.7554/eLife.27576.020
**Source data 6.** Individual data points *Figure 3—figure supplement 2*.
DOI: https://doi.org/10.7554/eLife.27576.021
**Figure supplement 1.** Two components analysis for the diffeomorphic scrambling experiment.
DOI: https://doi.org/10.7554/eLife.27576.014
**Figure supplement 2.** Shape sensitivity for dorsal left and dorsal right and ventral left and ventral right.
DOI: https://doi.org/10.7554/eLife.27576.015

*Figure 1*]. Notably, an interaction between pathway and hemisphere [$F_{(1,20)}$=10.85, p=0.003, $\eta_p^2$ = 0.35] was found and was not modulated by experiment [$F_{(1,20)}$<1]. This interaction was the result of a greater difference between the two pathways in the left than right hemisphere, and therefore further implicates hemispheric differences in shape processing.

Finally, the ROI analysis (*Figure 3—figure supplement 2*) confirms that the large-scale organization of object processing was largely replicated in the diffeomorphic scrambling experiment. In particular, in the dorsal pathway, shape sensitivity was first evident in V3b, reached a peak at IPS0/V3b and then decreased, and was not evident in the more anterior IPS4 and aIPS, confirming the posterior-anterior representational gradient. Note that the ROIs-based analysis did not permit an exploration of the ventral pathway second component, since this component was found to be in more anterior parts of the ventral pathway in the diffeomorphic scrambling experiment and the probabilistic atlas does not include these more anterior regions.

## Representational similarity analysis
### Box scrambling experiment
So far, we have presented results based on a univariate approach. Despite the robustness of this method and the convergence of the voxel-wise and ROI approaches, these analyses only consider the magnitude but not the pattern of the fMRI activation. Recent investigations employing multivariate approaches have successfully uncovered important features of the neural representations of objects in both the dorsal and the ventral pathways (e.g., *Bracci et al., 2017*; *Bracci and Op de Beeck, 2016*; *Fabbri et al., 2016*).

To extend our results, we utilized a multivariate approach of representational similarity analysis (*Kriegeskorte et al., 2008*; RSA). Notably, most previous investigations utilize the RSA approach to uncover the representational space of individual exemplars. The RSA employed here, however, is unlike the standard RSA procedure. Given that the present study used a block design to increase the statistical power in the service of the voxel-by-voxel univariate mapping described above, we

could not evaluate the representation of individual exemplars. Instead, the RSA was applied at the level of a block (where a single block contains intact or scrambled objects to differing degrees). This procedure still allows us to explore how the availability of shape information modulated the representational content of different ROIs.

The RSA computed the similarity of spatial activation patterns across conditions for different ROIs and then tested whether these data could be accounted for by a model in which representations of images will be more similar to each other if they possess similar levels of shape information. For example, in shape selective regions, the response pattern to intact images is predicted to be more correlated with the response pattern to images scrambled to four pieces compared with the response pattern to images scrambled to 64 pieces. Correspondingly, response patterns for the 256 scrambled inputs should be more correlated with voxels that respond to the 64 scrambled inputs versus intact objects (*Figure 4A*).

The results of the RSA are depicted in *Figure 4B*. Both early visual cortices and object-selective cortices were correlated with the shape model. The correlation of early visual cortices to the shape model can be explained by the negative slope observed in those regions that carry information about the presence/absence of a shape and about low-level similarities (*Bracci and Op de Beeck, 2016*). More importantly, and compatible with the univariate analysis, mid-posterior parts of the dorsal pathway and ventral pathways were highly correlated with the shape sensitivity model, and this sensitivity decreased in more anterior parts of the parietal cortex and temporal lobe.

Finally, RSA also permits a second-level analysis in which multidimensional scaling is used to place ROIs into a representational similarity space (here two-dimensional) such that those that are similar in representational structure are in closer proximity and those in which representational structure is dissimilar are more distant (*Figure 4B*) (see Materials and methods). This visualization further demonstrates the separation between early visual cortices and object-selective cortices. More importantly, this analysis also reveals the association in the representational structures of the two pathways. In particular, the patterns of activation in the posterior parts of the dorsal pathway (i.e. V3a-IPS1) were highly correlated with regions of the lateral ventral pathway (i.e., LO1, LO2, and TO1).

To quantify the representational similarity within and between pathways, we compared the correlations of the object-selective ROIs within each pathway (i.e. lateral-ventral ROIs to inferior-ventral ROIs and posterior-dorsal ROIs to anterior-dorsal ROIs) to the correlations of the object-selective ROIs between the two pathways (i.e., posterior-dorsal ROIs with lateral-ventral ROIs). Interestingly, this analysis revealed that the between-pathways correlations were higher than the within-pathway correlations both in the ventral [$F_{(1,10)}$=26.1, p=0.0004, $\eta_p^2$ = 0.72,, CI{0.05, 0.13}] and dorsal [$F_{(1,10)}$=19.3, p=0.001, $\eta_p^2$ = 0.65, CI{0.05, 0.16}] pathways, further challenging a binary distinction between the two pathways. Notably, this effect is not the result of the assignment of a particular ROI to the posterior/anterior dorsal pathway, and was replicated even when aIPS was excluded from the analysis, or when IPS2 was included in the posterior (rather than anterior) dorsal pathway.

The RSA results converge with the univariate results and demonstrate that regions in the dorsal and ventral pathway share highly similar representational structures based on the availability of shape information, thereby challenging the binary distinction between the two pathways.

## Diffeomorphic scrambling experiment

The results of the RSA are depicted in *Figure 5B*. Both early visual cortices and object-selective cortices were correlated with the shape model, however, these correlations were significantly reduced in comparison to the box scrambling experiment (for a direct comparison see *Figure 5—figure supplement 1*). Importantly, this reduction is not an artifact of noise in the data as evident from the noise ceiling calculation (gray bars, *Figure 5B*). The reduction in the correlation to the shape model is therefore more likely related to the fact that the impact of shape distortion was reduced for the diffeomorphic stimuli. Importantly however, the large-scale organization of the dorsal pathway (particularly in the right hemisphere) was replicated in the diffeomorphic scrambling experiment, as the highest correlations with the shape model were observed in V3a-b and IPS0.

A robust reduction in the correlation to the shape model was observed along the ventral pathway, particularly along its lateral surface. This was confirmed by an ANOVA that revealed an interaction between experiment and ROI [$F_{(10,200)}$ = 14.48, p=0.0000001, $\eta_p^2$ =.42]. The robust decrease of

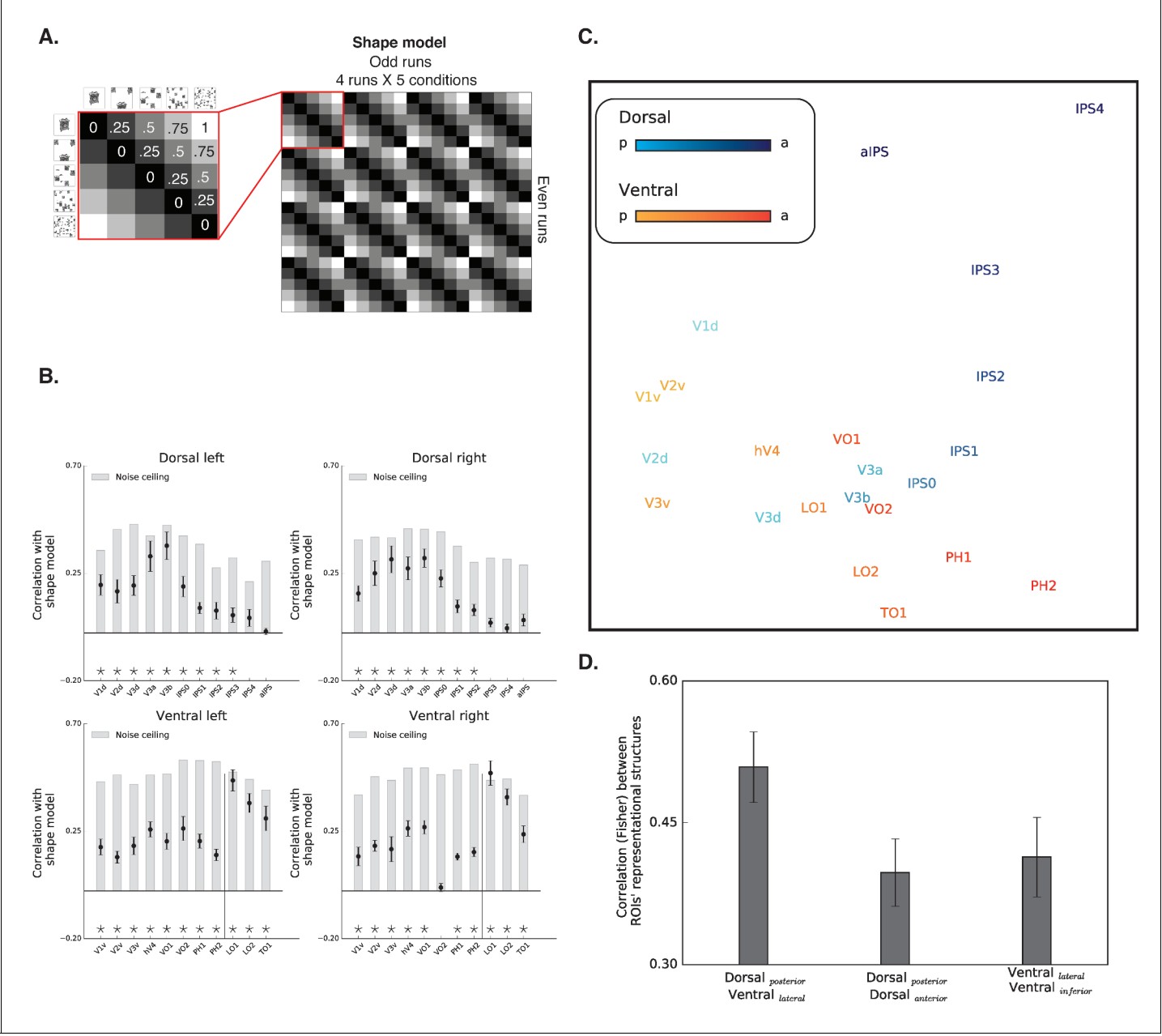

**Figure 4.** Box scrambling experiment - RSA analysis. (**A**) The shape model is a dissimilarity matrix which asserts that representations of blocks of images will be more similar to each other (i.e. lower dissimilarity) if they possess similar levels of shape information. The analysis correlates the pattern of activation from the odd (columns) and even (rows) runs. For clarity, the left matrix magnifies a part of the model, and the values reflect the predicted dissimilarity values (range from 0 – maximal similarity to 1 maximal dissimilarity). (**B**) Correlation with the shape sensitivity model is plotted as a function of ROI, defined from atlases, separately for each pathway and hemisphere. Black asterisks along the x-axis signify that a ROI is significantly correlated with the shape model (q < 0.05). The black vertical line separates the lateral and inferior ROIs of the ventral pathway. The correlation with the shape model was evident in the early visual cortices, as well as in object selective cortices. The correlation with the shape model reached a peak in the LO (ventral) and V3b (dorsal) and then decreased gradually. For each ROI, the bright gray bars reflect the noise ceiling (the reliability of the correlational patterns in each ROI, which approximates the upper limit of the correlations between the fMRI and shape model given the inherent noise in the data (see methods)). (**C**) MDS plot performed on the second-order correlation across ROI's (averaged across participant) revealed that posterior dorsal regions and lateral ventral regions had highly similar representational structure and that differentiation between the pathways emerged in more anterior parts of the two pathways. Darkness of the markers signify their location on the posterior (bright)-anterior (dark) axis. For clarity, ROIs for the left hemisphere are presented, but a highly similar MDS plot is obtained for the right hemisphere ROIs. (**D**) Statistical quantification of the second-order correlation reveals that correlation between posterior dorsal ROIs and lateral ventral ROIs were higher than the within-pathways correlations.
DOI: https://doi.org/10.7554/eLife.27576.022

*Figure 4 continued on next page*

Figure 4 continued

The following source data is available for figure 4:

Source data 1. Individual data points for *Figure 4B*.
DOI: https://doi.org/10.7554/eLife.27576.023
Source data 2. Individual data points for *Figure 4D*.
DOI: https://doi.org/10.7554/eLife.27576.024

correlation to the shape model, further suggests that the LOC codes shape information rather than a high-level abstraction of the object's identity.

The second-level RSA was compatible with that observed for the box scrambling experiment, reinforcing the notion that the large-scale organization of the two pathways was reliable and stable across the two experiments. The MDS plot shows that the patterns of activation in the posterior parts of the dorsal pathway (i.e. V3a-IPS1) were highly correlated with regions of the ventral pathway (e.g., LO), while more anterior parts of the two pathways had distinctive representational structures. The statistical quantification replicated the box scrambling experiment findings, with higher correlations of the between-pathways ROIs (lateral-ventral and posterior-dorsal) compared with the correlations of the within-pathways ROIs [within ventral: $F_{(1,10)}$ = 54, p=0.00002, $\eta_p^2$ = 0.84,, CI{0.08, 0.15}; within dorsal: $F_{(1,10)}$ = 130, p=0.000001, $\eta_p^2$ = 0.94, CI{0.11, 0.17}]. Notably, as in the box scrambling experiment, this effect was replicated even when aIPS was excluded from the analysis, or when IPS2 was included in the posterior (rather than anterior) dorsal pathway.

## Correlation between fMRI activation and object recognition performance

A key question that emerges is whether the neural patterns we have uncovered bear any relation to perceptual performance. Previous studies have observed a robust correlation between fMRI activation in object-selective regions of the ventral pathway and object recognition abilities (*Avidan et al., 2002*; *Grill-Spector et al., 2000*). Because this correlation has not been examined for the dorsal pathway in the context of non-spatial perceptual tasks (rather than, for example, working memory, *Jeong and Xu, 2016*), we correlated the fMRI activation for the different levels of scrambling with recognition abilities measured outside the scanner.

### Box scrambling experiment

Object recognition abilities decreased as a function of scrambling, as revealed by a main effect of scrambling level in the repeated measures ANOVA [$F_{(1,10)}$ = 477, p=0.000001, $\eta_p^2$ = 0.97], and a robust simple effect of the linear contrast [$F_{(1,10)}$ = 1634, p=0.000001] (*Figure 5A*, black line). Given this linear relationship, and a similar linear relationship between fMRI signal and scrambling level, any correlation between the fMRI signal and behavioral performance might be attributed to the shared correlation of these variables with the level of scrambling. Therefore, we computed the partial correlations between fMRI signal and behavioral performance of each participant after regressing out the shared correlation with the level of scrambling. Consistent with previous reports, fMRI activation along the ventral pathway was correlated with perceptual performance on both the inferior and lateral surfaces (r > 0, q < 0.05; *Figure 6B*, black markers). The more novel finding is that dorsal-pathway activation was also correlated with perceptual behavior in multiple ROIs along the IPS (r > 0, q < 0.05; *Figure 6B*), suggesting a functional role for the dorsal pathway in visual perception.

### Diffeomorphic scrambling experiment

In the diffeomorphic scrambling experiment, object recognition abilities decreased as a function of scrambling, as revealed by a main effect of level of scrambling in the repeated measures ANOVA [$F_{(1,9)}$=929, p=0.0000001, $\eta_p^2$ = 0.99], and a robust simple effect of the linear contrast [$F_{(1,9)}$=20596, p=0.0000001] (*Figure 6A*, gray line). Recognition performance for the scrambled versions (levels S2, S3 and S4) observed in the diffeomorphic scrambling were lower than those obtained for the box scrambling experiment, as evident from the interaction between experiment and level of scrambling [$F_{(4,76)}$=71, p=0.0000001, $\eta_p^2$ = 0.78] and the planned comparisons [$Fs_{(1,19)}$ > 5.7, p<0.05]. Hence, the lower fMRI sensitivity to object shape (i.e. flatter slope) in the diffeomorphic scrambling

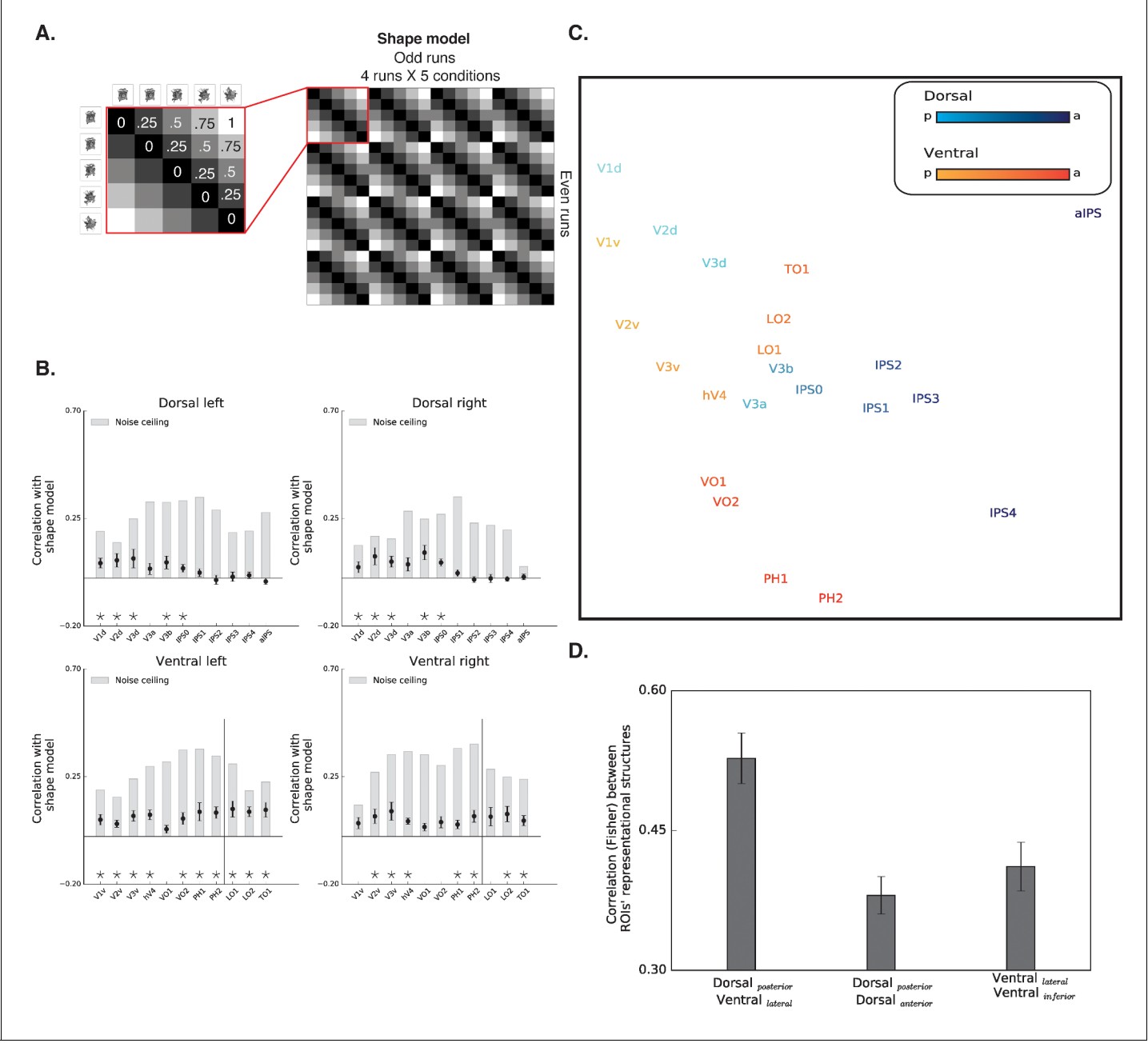

**Figure 5.** Diffeomorphic scrambling experiment - RSA analysis. (**A**) The shape model is a dissimilarity matrix which proposes more similar representations (i.e. lower dissimilarity) to blocks of images containing similar levels of shape information. For more details, see *Figure 4* legend. (**B**) Correlation with the shape sensitivity model is plotted as a function of ROI defined from atlases, separately for each pathway and hemisphere. Black asterisks signify that a ROI is significantly correlated with the shape model (q < 0.05). The black vertical line separates the lateral and inferior ROIs of the ventral pathway. Correlation with the shape model was evident in the early visual cortices, as well as in object selective cortices. Despite a reduction in the magnitude of the correlations in comparison to the box scrambling experiment, a similar pattern of sensitivity to shape information was observed. For each ROI, the bright gray bars reflect the reliability of the correlational patterns in each ROI, which approximates the upper limit of the correlations between the fMRI and shape model given the inherent noise in the data (see methods). (**C**) MDS plot performed on the second-order correlation across ROI's (averaged across participant) revealed that posterior dorsal regions and lateral ventral regions had highly similar representational structure and that differentiation between pathways emerged in more anterior parts of the two pathways. Darkness of the markers signify their location on the posterior (bright)-anterior (dark) axis. For clarity ROIs for the left hemisphere are presented, but a highly similar MDS plot is obtained for the right hemisphere ROIs. (**D**) Statistical quantification of the second-order correlation reveals that correlation between posterior dorsal ROIs and lateral ventral ROIs were higher than the within-pathways correlations. *Figure 5—figure supplement 1* shows the direct comparison between the two experiments. Black and gray asterisks signify that a ROI is significantly correlated with the shape model in the box scrambling experiment and the diffeomorphic

*Figure 5 continued on next page*

*Figure 5 continued*

scrambling experiment., correspondingly. Gray filled circles (q < 0.05) and gray filled triangles (q < 0.1) signify that, in a particular ROI, the correlation with the shape model was greater in the box scrambling experiment compared with the diffeomorphic scrambling experiment.

DOI: https://doi.org/10.7554/eLife.27576.025

The following source data and figure supplement are available for figure 5:

**Source data 1.** Individual data points for *Figure 5B*.
DOI: https://doi.org/10.7554/eLife.27576.027
**Source data 2.** Individual data points for *Figure 5D*.
DOI: https://doi.org/10.7554/eLife.27576.028
**Figure supplement 1.** Correlation with shape model for dorsal left and dorsal right and ventral left and ventral right.
DOI: https://doi.org/10.7554/eLife.27576.026

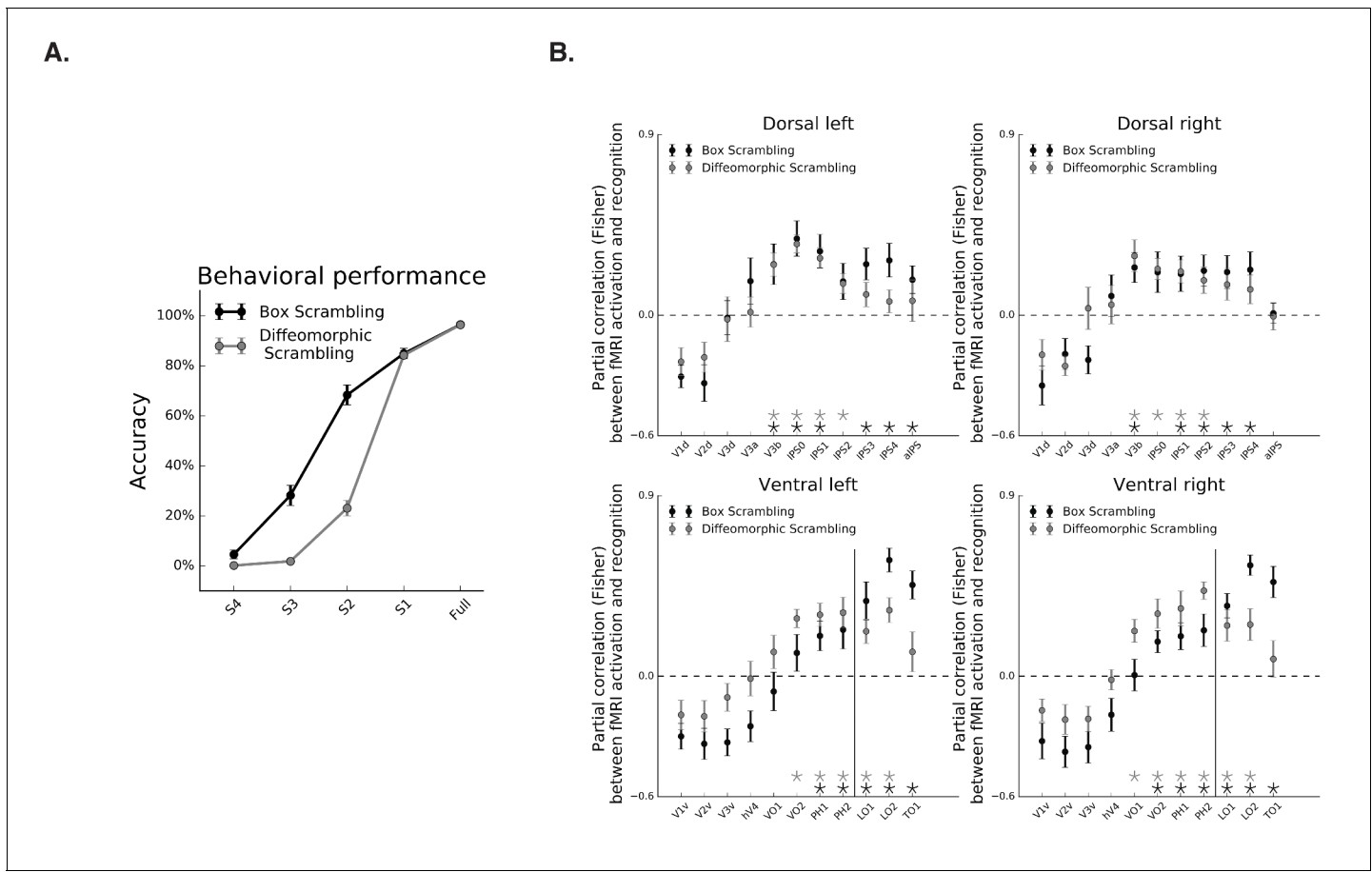

**Figure 6.** Correspondence between fMRI and object recognition performance. (**A**) Mean accuracy of recognition, obtained outside the scanner in the box scrambling experiment and the diffeomorphic scrambling experiment., as a function of scrambling. Recognition ability decreased as a function of scrambling. (**B**) Partial correlation between fMRI activation and recognition performance along the two pathways. Black and gray asterisks signify that a ROI evinces a significantly positive correlation between these two variables (r > 0, q < 0.05) in the box scrambling experiment and diffeomorphic scrambling experiment, respectively. The black vertical line separates the lateral and inferior ROIs of the ventral pathway. In both experiments, object recognition abilities were correlated with fMRI responses across different ROIs in the mid-anterior parts of the ventral and dorsal pathways.
DOI: https://doi.org/10.7554/eLife.27576.029
The following source data is available for figure 6:

**Source data 1.** Individual data points for *Figure 6A*.
DOI: https://doi.org/10.7554/eLife.27576.030
**Source data 2.** Individual data points for *Figure 6B*.
DOI: https://doi.org/10.7554/eLife.27576.031

experiment could not be related to superior recognizability of the distorted images in this experiment. The comparison between the behavioral performances in the two experiments should be interpreted with caution, however, since participants in the box scrambling experiment were exposed to each object in all levels of scrambling whereas, in the diffeomorphic scrambling experiment, participants were exposed to a particular object only in one level of distortion.

As in the box scrambling experiment, the partial correlation between performance and fMRI signal, with the level of scrambling as a covariate, revealed a correlation between fMRI activation and perceptual performance on both the inferior and lateral surfaces of the ventral pathway (r > 0, q < 0.05; *Figure 6B*. gray markers). Importantly, in this analysis, the coupling between behavioral performance and fMRI activity was also apparent in different ROIs along the IPS (r > 0, q < 0.05.; *Figure 6B*. gray markers).

### Effects of visuomotor cues

The dorsal pathway is known to be highly responsive to images of tools, which convey visuomotor information (*Chao and Martin, 2000*; *Lewis, 2006*; *Mruczek et al., 2013*). Our stimulus set included pictures of both tools and non-tool objects, and it could be argued that the shape sensitivity along the dorsal pathway may be a product of visuomotor associations related specifically to tools rather than of more general shape processing.

Closer scrutiny of the image statistics suggests that a direct comparison between tools and objects is ill-advised in the present study, because the image statistics from the two categories differed dramatically (e.g., number of pixels: objects [mean = 50,768, SD = 15,665] and tools [mean = 17,475, SD = 10,158]) and shape attributes (e.g., tools are usually elongated, while objects have more diverse shapes) (*Bracci and Op de Beeck, 2016*; *Chen et al., 2017*). Nevertheless, to ensure that the activation in the dorsal pathway was not solely a result of visuomotor associations, we reanalyzed the data of the two experiments using only the fMRI blocks in which objects (and not tools) were presented. Notwithstanding the loss of statistical power, the reanalysis fully reproduced the analyses described above at the voxel-wise and ROI level (see *Figure 2—figure supplement 1* upper panel, *Figure 3—figure supplement 1*-upper panel). The results of the two experiments, therefore, cannot be ascribed to the presence of tool stimuli that convey visuomotor associations, and, instead, bolster to the conclusion that posterior regions of dorsal cortex are responsive to the shape of the visual input.

## Discussion

In light of recent findings indicating that the neural representation of object shape may be mediated not only by the ventral but also by the dorsal visual pathway, the present study was designed to characterize and compare the large-scale organization of shape processing across the two visual pathways. Three key findings were obtained. First, the two pathways followed a similar topographical organization with a posterior to anterior initial increase and then decrease in shape sensitivity. Second, the representational structures of regions in the dorsal and ventral pathways were highly correlated with each other. Third, the activation of posterior dorsal pathway regions, which were sensitive to shape, were correlated with recognition performance, suggesting that these regions might contribute to perceptual behaviors. Together, these findings challenge the binary distinction between the two pathways.

### Topographical organization of shape processing

Consistent with early investigations (e.g., *Grill-Spector et al., 1998*; *Grill-Spector and Weiner, 2014*; *Lerner et al., 2001*; *Murray et al., 2002*), shape sensitivity, was not evident in the caudal parts of the ventral pathway (i.e., early visual cortex) and emerged in the rostral and lateral parts of the ventral pathway, reflecting the hierarchical nature of object processing. However, in the transition from high-level visual regions (i.e., parahippocampal gyrus, fusiform gyrus) to even more anterior temporal regions, shape sensitivity was still present, albeit reduced (i.e., a flatter slope). Interestingly, in recent years, accumulating evidence has shown that the anterior and medial regions of the temporal lobe (i.e., perirhinal and entorhinal cortices) may also play a role in object recognition (*Behrmann et al., 2016*); for a review see *Murray et al., 2007*). The results of the present investigation are compatible with the view that these more anterior representations may be confined to

high-level object properties (such as conjunction of multiple features see, *Barense et al., 2007*; or familiarity, see *Martin et al., 2013*) and to memory-based representations, and less to shape and geometric information per se.

The novel findings pertain more to the nature of shape processing along the dorsal pathway. In particular, shape sensitivity increased from early visual cortex to extrastriate cortex, reached its peak sensitivity in the posterior parietal cortex and then gradually decreased in more anterior regions, closer to the central sulcus. These findings extend previous studies in both humans and non-human primates that have demonstrated that, similar to the ventral pathway, the dorsal pathway derives object representations. (e.g., *Denys et al., 2004*; *Freud et al., 2015*; *Konen and Kastner, 2008*; *Theys et al., 2015*; *Van Dromme et al., 2016*).

The similarity between the two pathways suggest that the topographical organization is constrained by similar factors, namely cortical distance from the visual cortex and connectivity to other cortical systems (i.e., the motor system in the dorsal pathway and semantic system in the ventral pathway).

Finally, the similarity in the large-scale organization of the two pathways was further confirmed by an RSA approach that revealed that the representational structure of regions in the posterior dorsal pathway was highly correlated with the representational structure of lateral-ventral ROI. Notably, despite these similarities, differences were also observed between the two pathways: shape sensitivity was greater in the ventral than dorsal pathway, reflecting the centrality of this pathway in object perception, and shape sensitivity reached its maximum values in more anterior regions of the ventral than dorsal pathway.

## What aspects of object properties are encoded by the dorsal pathway?

Some fMRI investigations have provided evidence that pictures of tools activate regions in the dorsal pathway (*Chao and Martin, 2000*; *Valyear et al., 2007*), even when no overt visuomotor task was required. Nevertheless, one might speculate that the observed activation might still reflect visuomotor plans that are associated with the tool being displayed. More recent studies, however, have documented dorsal activation for both 2D and 3D images, even when the stimuli do not have any visuomotor association and the task is not action-based (*Freud et al., 2015*; *Konen and Kastner, 2008*; *Zachariou et al., 2014*). Moreover, when BOLD responses to tools were compared to non-tools (but graspable) objects, no difference was observed between the two categories in the posterior parts of the dorsal pathway, but greater activation was observed for tools than for objects in more anterior regions (*Mruczek et al., 2013*). In the current work, shape sensitivity within the dorsal pathway was found both for tools and non-tool objects. Together, these results suggest that the representations subserved by the posterior regions of the dorsal pathway are not limited to visuomotor associations evoked by the object. Instead, the neural representation may reflect the processing of different shape cues such as the 3D status of the object (*Berryhill et al., 2009*; *Freud et al., 2017a*; *Konen and Kastner, 2008*; *Van Dromme et al., 2016*) and/or object elongation (*Chen et al., 2017*; *Fabbri et al., 2016*).

Recent studies have used multivariate analytic approaches to elucidate the visual and cognitive features represented by the dorsal (and ventral) regions. For example, *Bracci and Op de Beeck, 2016* mapped the cortical sensitivity for a stimulus set in which shape and category (e.g., animals, musical instruments, tools) were dissociated. Similar to the present findings, posterior LOC was more correlated with a shape model than a category model. In the dorsal pathway, the TOS, which corresponds to posterior ROIs in the probabilistic atlas used here, was also highly correlated with the shape model, and not with the category model. Nevertheless, in this previous investigation, some anterior dorsal ROIs were more correlated with the category model, than the shape model. As discussed below, dorsal pathway representations were found to be highly sensitive to task properties (*Bracci et al., 2017*) and therefore, the discrepancies between the present investigation and Bracci and Op de Beeck's findings might be related to the nature of the task: while Bracci and Op de Beeck asked participants to compare the real-size of preceding images (encouraging object-based processing), in the present investigation, we deliberately avoided explicit processing of the stimuli and had participants complete an orthogonal fixation-based task.

The comparison between the box scrambling experiment and the diffeomorphic scrambling provided additional information on the nature of shape representations derived by the dorsal pathway. The diffeomorphic transformation distorted the object's identity, while preserving the presence of

some shape information (to a greater degree than was true of the box scrambling manipulation). In contrast to the inferior surface of the ventral pathway, along the dorsal pathway a decrease in shape sensitivity was found for this manipulation, compared with the results obtained from the box scrambling experiment. Hence, our results suggest that dorsal pathway representations are tied to the presence of a single coherent shape, even if it is not identifiable as a familiar object and lacks important visual properties. The residual perceptual abilities of patients with visual agnosia are consistent with such interpretation. In particular, the dissociable dorsal representations can support sensitivity to particular attributes (such as the 3D structure of an object) but cannot support intact recognition abilities (*Freud et al., 2017a*).

Note that this interpretation ought to be considered speculative as the shape attribute is confounded with other visual properties that constitute a shape in the diffeomorphic scrambling experiment: our image analysis procedures revealed that a host of factors, related to low-, mid- and high-level shape cues, such as defined perimeter, entropy and homogeneous texture, were also better preserved for the diffeomorphic than box-scrambled stimuli. Moreover, and in contrast to previous investigations, the RSA in the present study was based on a block design experiments, and therefore, could not uncover the representational content of a specific exemplar. Future studies should, therefore, explore the importance of different visual properties to dorsal pathway representations, and, by doing so, evaluate whether dorsal and ventral representations differ quantitatively (or also qualitatively).

## Functional contribution of dorsal pathway representations to perception

According to *Goodale and Milner (1992)*, the visual pathways should be described in terms of their functions, rather than in terms of the visual information that is represented. The question then is, what behavioral functions are subserved by dorsal pathway representations? Here, we provide novel evidence for a correlation between recognition abilities and fMRI shape sensitivity in the posterior part of the dorsal pathway, even when the correlation of these variables with scrambling levels was partialled out. Despite the inability to infer causality from those correlations, they extend seminal findings that found correlations between perceptual performance and fMRI activation in different regions of the ventral pathway (*Grill-Spector et al., 2000*) and point to a plausible functional contribution of the dorsal pathway to object perception.

Consistent with this suggestion, a patient with an occipitoparietal lesion was impaired in the perception of objects defined by monocular and binocular cues (*Berryhill et al., 2009*). Moreover, recent studies, using TMS in humans and reversible deactivation in non-human primates, have successfully established a causal relationship between the dorsal activation and perceptual classification (*Van Dromme et al., 2016*; *Zachariou et al., 2017*).

Notably, as evident from visual agnosia patients and from the results of the diffeomorphic scrambling experiment, the dorsal pathway appears not to be *sufficient* to support intact perception and object recognition, and the ventral pathway remains more central to this function. Even so, future studies should assess the contribution of dorsal pathway representations to different perceptual tasks in humans. This research would help elucidate the functional role of the dorsal pathway representations in object perception.

## Task demands might modulate object representations in the dorsal pathway

In both experiments reported here, there was a decrease in shape sensitivity in the anterior parts of the dorsal pathway, in both univariate and multivariate analyses. This was particularly evident in the aIPS, a region associated with computations related to the representations of objects in the context of visuomotor tasks (*Culham et al., 2003*; for a recent review see, *Gallivan and Culham, 2015*). However, several other neuroimaging investigations in humans and electrophysiological studies non-human primates have revealed shape sensitivity in the anterior parts of the IPS (*Durand et al., 2007*; *Freud et al., 2017b*; *Janssen et al., 2000*; *Orban, 2011*; *Theys et al., 2013*; *Theys et al., 2015*). This apparent contradiction raises the possibility that the nature of the task modulates shape sensitivity in the dorsal pathway, and, indeed, a recent study found that the aIPS coded object shape and elongation during grasping but not during passive viewing (*Fabbri et al., 2016*). Moreover, task

demands were found to modulate the nature of representations in other regions of the dorsal pathway, under perceptual (rather than visuomotor) tasks, further suggesting that dorsal pathway representations might be sensitive to tasks demands (*Bracci et al., 2017*). Notably, in contrast to these findings, a recent study revealed that the ventral pathway is less affected by task demands (*Bugatus et al., 2017*), revealing potential differences in the way task modulates the representations in the two pathways.

## Conclusion

The present study uncovered novel evidence for the nature of shape representations along the dorsal pathway and its similarity to representations along the ventral pathway. In two fMRI experiments, using a variety of analytical approaches, we found that posterior extrastriate regions of the dorsal pathway are highly sensitive to shape information (but less to object identity), that the magnitude of their activation is correlated with perceptual behaviors and that in some regions, the representations derived are highly similar to those in some regions of ventral cortex. This sensitivity decreases in anterior regions of the dorsal pathway reflecting a gradual shift from representation-for-perception to representation-for action, leading to the conclusion that there is a representational continuum from more posterior areas tuned to visual properties of the objects in the input to more anterior areas tuned to motor aspects of the observed objects.

# Materials and methods

## Participants

Twenty-two right-handed participants (box scrambling experiment: Eleven participants, nine males; mean age: 31, range: 19–46 years, diffeomorphic scrambling experiment: Eleven participants, five males; mean age: 25, range: 19–46 years). The data obtained from two additional participants were not analyzed due to excessive head movements (>3 mm) during multiple scans. One additional participant (diffeomorphic scrambling experiment) did not complete the behavioral session, and therefore, for this experiment, the correlation between behavioral performance and fMRI signal was calculated based on the results of ten participants. All participants had normal or corrected-to-normal vision and were financially compensated for their participation. Informed consent was obtained prior to the study. All experimental procedures were approved by the Institutional Review Board of Carnegie Mellon University.

## Stimuli

### Box scrambling experiment

Stimuli were 160 grayscale pictures of everyday objects (80 pictures) and tools (80 pictures) downloaded from The Bank of Standardized Stimuli (BOSS) (*Brodeur et al., 2010*; *Brodeur et al., 2014*). Each image was divided into 4, 16, 64 and 256 squares that were randomly rearranged, resulting in five levels of scrambling (*Figure 1*) (most scrambled to intact - S4, S3, S2, S1 and Full). Each version of each stimulus was presented twice in the experiment.

### Diffeomorphic scrambling experiment

Stimuli were 320 grayscale pictures of everyday objects (160 pictures) and tools (160 pictures) downloaded from The Bank of Standardized Stimuli (BOSS) (*Brodeur et al., 2010*; *2014*). In contrast to the box scrambling experiment, only one version of each picture was presented to a particular participant (counterbalanced across participants) to ensure that sensitivity to object shape was not modulated by within-experiment priming or adaptation. For each object, five versions were created (no distortion, one distortion step, two distortion steps, four distortion steps and eight distortion steps, distortion level = 80; Matlab function was provided by Stojanoski and Cusack, *Figure 1*). Each selected version of a picture was presented four times throughout the experiment.

### Comparison between stimuli as function of distortion method

To quantify 'the goodness of shape properties' of the stimuli used in the two experiments, we utilized a set of algorithms that measured low-, mid- or high-level shape attributes. To assess low-level

similarity, we used a pixel-similarity analysis (*Op de Beeck et al., 2008*) to compare the pixels for each distorted image relative to the intact version of the image. The pixel similarity score was computed as follows and ranged from 0 (completely dissimilar) to 1 (similar):

$$Pixel\ similarity = 1 - \frac{\sqrt{\sum_1^n (P_{n1} - P_{n2})^2}}{n}$$

where n is the number of pixels, $P_{n1}$ is a given pixel in the intact image and $P_{n2}$ is a given pixel in the distorted image. This analysis revealed greater similarity (higher values) between the intact images and distorted images in diffeomorphic scrambling experiment compared with box scrambling experiment (box scrambling: 0.7 ± 0.02; diffeomorphic scrambling: 0.87 ± 0.03).

To characterize mid-level attributes, we measured the image entropy (Matlab command: entropy) which provides an estimation of the overall disorder in the image with greater values reflect greater entropy. In addition, image homogeneity (Matlab command: graycoprops) was analyzed to examine the extent to which image texture was altered by the two scrambling methods. For this measure, lower values reflect reduced homogeneity and less uniform texture. These analyses showed greater entropy and lower homogeneity for the box scrambling stimuli (Image entropy: box scrambling: 1.07 ± 0.07; diffeomorphic scrambling: 1.02 ± 0.13; Image homogeneity: box scrambling: 0.98 ± 0.007; diffeomorphic scrambling: 0.99 ± 0.005)

Finally, to examine the distortion of high-level shape information (i.e., closed boundaries, continuous counter) we defined the edges of each image by detecting all pixels adjacent to a background (i.e., white) pixel, and then calculated the convex hull, which is the smallest convex set that contains a particular image (*Andrew, 1979*). Next, for each edge pixel, the distance to the nearest convex hull pixel was computed. The sum of all distances for each image was normalized by dividing it by the sum of distances for the intact version of that image. Thus, greater values of the normalized distance reflect increased shape complexity relatively to the intact image. As with the other algorithms, this procedure confirmed that shape properties were distorted even by the diffeomorphic images, but were still better preserved for these images compared with the box-scrambled images across all levels of scrambling (see *Figure 1B*).

## Procedure

### fMRI

Stimuli were presented in a pseudorandomized order using the E-Prime 2.0 software (Psychology Software Tools, Inc., Pittsburgh, PA, USA) and projected via a liquid crystal display (LCD) screen located at the back of the scanner bore, behind the subject's head. Stimuli were presented within a square frame (visual angle of 4.5° X 4.5°) on a white background. Participants viewed the stimuli through a tilted double-mirror setup mounted above their eyes on the head coil. Prior to scanning, they all completed a short training session with the experimental tasks and stimuli.

### MRI setup

Participants were scanned in a Siemens Verio 3-Tesla magnetic resonance imaging scanner with a 32-channels coil at Carnegie Mellon University. A structural scan was acquired using a T1-weighted protocol that included 176 sagittal slices (1 mm thickness, in-plane resolution = 1 mm, matrix = 256×256, repetition time = 2300 ms, echo time = 1.97 ms, inversion time = 900 ms, flip angle = 9 °). Functional images based on the blood oxygenation level-dependent (BOLD) signal were acquired with a gradient-echo, echoplanar imaging sequence (TR = 1.5 s, TE = 30 ms, flip angle 73 °). To achieve full coverage of both visual pathways, 43 axial slices (slice thickness = 3 mm, gap = 0 mm, in-plane resolution = 3 mm) were acquired in eight runs of 227 volumes each (such that each run lasted 5 min, 40.5 s).

### Object scrambling experiment

No participant completed more than one experiment. In each of eight runs in each experiment, participants viewed pictures that were blocked by the five levels of scrambling (S4, S3, S2, S1 and Full). After an initial fixation of 10.5 s, twenty 9 s blocks, each comprised of ten stimuli displayed for 600 ms followed by 300 ms fixation, were presented. The blocks were separated by 7.5 s fixation periods. Participants were instructed to fixate on the cross in the center of the display. To maintain

attention throughout the scan, participants were required to indicate, via a button press, when the color of the fixation cross changed from black to red. There were one or two fixation color changes per block of ten stimuli.

### Behavioral object recognition test

Participants returned 2–3 weeks after completing the fMRI experiment. Seated 50 cm in front of a computer screen in a darkened room, they were shown the same stimuli they had viewed in the scanner and were instructed to name aloud each stimulus. The experimenter tracked the accuracy of their responses. Stimuli were presented in a pseudo-randomized fashion, for 600 ms (as in the fMRI experiment), with each picture presented once.

## Data analysis

fMRI raw data are available at https://doi.org/10.1184/R1/c.3889873.v1 and were processed using BrainVoyager 20.2 software (Brain Innovation, Maastricht, Netherlands; RRID:SCR_013057), MRIcron (RRID:SCR_002403), complementary in-house software written in Matlab (The MathWorks, Inc, Natick, MA, USA; RRID:SCR_001622; see source code) and *R Development Core Team (2009)*. Pre-processing included 3D-motion correction and filtering of low temporal frequencies (cutoff frequency of 2 cycles per run). No spatial smoothing was applied to allow the voxel-wise analysis. All scans were transformed to Montreal Neurological Institute (MNI) space (*Fonov et al., 2011*). Three main analytical approaches were employed: a novel voxel-wise approach that allows a continuous mapping of shape processing along the two pathways, a more traditional ROI analysis and a multivariate representational similarity analysis (RSA).

### Voxel-wise approach
#### Definition of a group-level mask

To map the topographical origination of shape processing along the two pathways in a continuous fashion, we generated a voxel-wise map of shape sensitivity of every visually selective voxel. A group-level mask of all visually selective voxels was generated by performing a random-effects general linear model (GLM) analysis across all 22 participants. This mask allowed us to compare the results of the two experiments directly, with the same voxels sampled in the two experiments. For each participant, the eight runs were concatenated and analyzed using a GLM, and beta value estimates of activation levels were calculated for each of the five conditions (scrambling level –S4, S3, S2, S1, Full). If a voxel was reliably responsive to any of these conditions ($q < 0.05$, false-discovery rate (FDR)-corrected, *Benjamini and Hochberg, 1995*) relative to fixation, it was included in the mask, which was then applied in the individual subject analysis. All voxels superior to the calcarine sulcus were considered part of the dorsal pathway, and all the voxels inferior to the calcarine sulcus were considered part of the ventral pathway. This approach yielded the following distribution: right dorsal pathway (941 functional voxels), left dorsal pathway (1155 functional voxels), right ventral pathway (1793 functional voxels) and left ventral pathway (1736 functional voxels).

#### Voxel-based piecewise regression analysis

For each participant, a voxel-wise GLM was conducted and a beta value for each of the five conditions was calculated. Next, to assess shape sensitivity, the slope of activation (beta values) as a function of scrambling level was calculated for each voxel. A positive slope reflects an increase in activation as the level of scrambling decreases (from S4 to intact), and therefore reflects greater shape sensitivity. A negative slope represents a decrease in activation as the level of scrambling decreases, and, as such, may reflect greater sensitivity to local elements and edges, which are more frequent in increasingly distorted images (see *Figure 2A*).

The location of a voxel on the posterior-anterior axis (i.e., the y-coordinate of each voxel) served as the independent variable, and shape sensitivity (i.e., slope) served as the dependent variable. Since an initial inspection showed that the topographical organization might not be linear, we conducted a piecewise regression analysis separately for each participant, pathway and hemisphere, using the shape language modelling (SLM) toolbox available in Matlab. Piecewise regression analysis allows the partitioning of the independent variable into multiple linear components, and then, for each component, a regression line is calculated. To avoid overfitting of the data, we limited the

number of segments to two, and the segmentation point was automatically derived to maximize the $R^2$. Notably, in each pathway, the piecewise linear regression robustly increased the $R^2$ was compared with a $R^2$ obtained from a simple linear regression (ps < 0.005, see *Table 1*), suggesting that the former provides a better fit to the data than a one-component linear model.

Note that because the y-axis (posterior-anterior) does not take into account the curvature of the brain, in a separate analysis, we computed the distance between each voxel and the most posterior voxel (using both Y and Z coordinates). We then, again, explored slope sensitivity per voxel as a function of this distance measure.

This analytical approach describes the spatial organization of shape processing in each visual pathway using two linear components. For each component, positive correlation values reflect an increase in shape sensitivity and negative correlation values reflect a decrease in shape sensitivity as a function of the posterior-anterior location of a voxel.

## ROI approach
### Definition of ROIs
As a means of offering converging evidence and benchmarking our findings to studies that have examined shape sensitivity as a function of ROI, we also analyzed our data per ROI. We defined ROIs based on a probabilistic atlas (*Wang et al., 2015*) that includes regions along the ventral and the dorsal visual pathways. We used the maximum probability map that identifies the most probable region for any given point. Note that IPS5, SPL, and TO2 included only a few voxels and, therefore, were not included in our analysis. The FEF is located in the frontal lobe, and, thus, was not included here. Since we were also interested in more anterior parts of the dorsal pathway, we defined the anterior intraparietal sulcus (aIPS) bilaterally by selecting voxels located in the vicinity of the junction of the intraparietal sulcus and the post-central sulcus based on coordinates extracted from Neurosynth (http://neurosynth.org/) using the search term 'Grasping' (*Hutchison and Gallivan, 2016*). Finally, since our stimuli were always presented at the center of the visual field, for early visual ROIs (V1v, V2v, V3v, V1d, V2d, V3d), we defined a more restricted ROI (5 mm sphere) at the posterior part of the original ROIs as defined in the probabilistic atlas, in order to capture foveal representation (*Wang et al., 2015*).

For each ROI, beta weights for each of the five conditions were calculated, and then, as with the voxel-wise analysis, the linear slope between the beta weights was extracted and served as a measure of shape sensitivity of that ROI. Note that the ventral mask used for the voxel-wise analysis extends to more anterior parts of the temporal lobe, compared with the ROIs available from the atlas, and therefore some differences between these two analytical approaches are expected.

### Representational similarity analysis (RSA)
The analyses thus far only consider beta weights and do not permit a more fine-grained analysis of the pattern of activation per ROI. To provide a more comprehensive understanding of the representational space in the different ROIs, we also conducted a multivariate investigation. To do so, beta weights from the different ROIs were extracted separately for each run and condition (8 runs x 5 conditions = 40) and the runs were separated into two sets with similar visual information contained in each set. The correlations between all the odd and even runs were extracted resulting in an asymmetric correlation matrix (20 × 20). To examine whether shape information was similarly encoded by different ROIs, we computed the Spearman correlation between the correlation matrix of each ROI and a shape model correlation matrix (see *Figures 4* and *5*) that proposes that representations of

**Table 1.** Comparison between simple regression and piecewise (two components) regression.

| | Simple correlation | | | | Two linear components | | | |
|---|---|---|---|---|---|---|---|---|
| | Ventral r | Ventral l | Dorsal r | Dorsal l | Ventral r | Ventral l | Dorsal r | Dorsal l |
| Box scrambling experiment | 0.04 ± 0.01 | 0.03 ± 0.01 | 0.05 ± 0.02 | 0.1±0.02 | 0.23 ± 0.03 | 0.26 ± 0.04 | 0.27 ± 0.03 | 0.22 ± 0.02 |
| Diffeomorphic scrambling experiment | 0.13 ± 0.02 | 0.11 ± 0.01 | 0.03 ± 0.01 | 0.03 ± 0.01 | 0.26 ± 0.03 | 0.33 ± 0.02 | 0.18 ± 0.03 | 0.17 ± 0.02 |

The mean $R^2$ (and standard error) based on a simple correlation or the two linear components regression that was utilized for the voxel-wise analysis.
DOI: https://doi.org/10.7554/eLife.27576.032

images will be more similar to each other if they possess similar levels of shape information. These correlation values were Fisher transformed to permit further statistical comparisons. One-sample t-test ($p < 0.05$, FDR corrected) was used to determine whether a specific ROI conveys shape information ($r > 0$). A series of t-test independent samples tests ($p < 0.05$, FDR corrected) were used to compare between the correlation obtained in the box scrambling experiment and the diffeomorphic scrambling experiment.

To estimate the upper limit of the correlation given the noise in the data, we calculated reliability for each ROI (*Bracci and Op de Beeck, 2016*). The correlation matrix of each subject in each ROI was correlated with the average matrix obtained from all other subjects. These values were averaged across participants and served as the reliability index. The reliability index is provided for reference (bright gray bars) in *Figures 4* and *5*.

Finally, the pattern of activation in each ROI was correlated with the activation pattern of all other ROIs. To visualize the similarity of the response patterns of the different ROIs, we averaged the correlation matrices of all the participants and utilized the Multi-Dimensional Scaling approach (MDS, 'mdscale' Matlab command). To statistically quantify the representational similarity between ROIs, for each participant, we averaged the correlation values between the lateral-ventral ROIs (LO1, LO2 and TO1) and inferior-ventral ROIs (VO1-PH2) (within ventral pathway), between posterior dorsal (V3a-IPS1) ROIs and anterior dorsal ROIs (IPS1-aIPS) (within dorsal pathway) and the correlations between the posterior-dorsal and lateral-ventral ROIs (between pathways). These values were then subject to a repeated-measures ANOVA.

### Correlation with object recognition

The univariate analyses described above included just five levels (scrambling), limiting the ability to calculate a reliable correlation between participants' BOLD response and object recognition abilities. We therefore divided the fMRI data into two datasets, in which different objects were presented, and we separated the data from the tools and objects categories. This procedure yielded 20 beta weights for each ROI. For the purpose of correlating the neural and behavioral data, we split the behavioral data into 20 subsets as was done with the neural data. Partial correlations between behavioral performance and fMRI signal were computed, controlling for the correlation of these two variables with the level of scrambling.

## Acknowledgements

The authors thank Bobby Stojanoski and Rhodri Cusack for providing the code for generating diffeomorphic transformations of the stimuli, and Rafael Malach and the VisCog group at CMU for fruitful discussion.

## Additional information

### Competing interests

Jody C Culham: Reviewing editor, *eLife*. The other authors declare that no competing interests exist.

### Funding

| Funder | Grant reference number | Author |
| --- | --- | --- |
| Israel Science Foundation | Grant No. 65/15 | Erez Freud |
| Yad Hanadiv Postdoctoral Fellowship | | Erez Freud |
| National Science Foundation | BCS-1354350 | David C Plaut Marlene Behrmann |
| Pennsylvania Department of Health | Commonwealth Universal Research Enhancement Program | David C Plaut |

| Canadian Institutes of Health Research | MOP 130345 | Jody C Culham |

The funders had no role in study design, data collection and interpretation, or the decision to submit the work for publication.

## Author contributions

Erez Freud, Conceptualization, Data curation, Formal analysis, Funding acquisition, Investigation, Visualization, Methodology, Writing—original draft, Writing—review and editing; Jody C Culham, Marlene Behrmann, Conceptualization, Supervision, Funding acquisition, Investigation, Methodology, Project administration, Writing—review and editing; David C Plaut, Conceptualization, Supervision, Funding acquisition, Investigation, Methodology, Writing—review and editing

## Author ORCIDs

Erez Freud  http://orcid.org/0000-0003-3758-3855
Jody C Culham  http://orcid.org/0000-0003-0754-2999
David C Plaut  http://orcid.org/0000-0002-0399-7120
Marlene Behrmann  http://orcid.org/0000-0002-3814-1015

## Ethics

Human subjects: As detailed in the manuscript, all subjects had normal or corrected-to-normal vision and were financially compensated for their participation. Informed consent and consent to publish was obtained in accordance with ethical standards set out by the Declaration of Helsinki (1964) and with procedures approved by the IRB committee of Carnegie Mellon University.

## Decision letter and Author response

Decision letter https://doi.org/10.7554/eLife.27576.038
Author response https://doi.org/10.7554/eLife.27576.039

# Additional files

## Supplementary files

• Source code 1. Matlab code for the voxelwise analysis.
DOI: https://doi.org/10.7554/eLife.27576.033
• Transparent reporting form
DOI: https://doi.org/10.7554/eLife.27576.034

## Major datasets

The following dataset was generated:

| Author(s) | Year | Dataset title | Dataset URL | Database, license, and accessibility information |
| --- | --- | --- | --- | --- |
| Freud E, Culham J, Plaut D, Behrmann M | 2017 | The large-scale organization of shape processing in the ventral and dorsal pathways | https://doi.org/10.1184/R1/c.3889873.v1 | Available at figshare under a CC0 Public Domain licence |

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
