## [Decision Letter]

Thank you for submitting your article "The large-scale organization of shape processing in the ventral and dorsal pathways" for consideration by *eLife*. Your article has been reviewed by two peer reviewers, and the evaluation has been overseen by Nick Turk-Browne as the Reviewing Editor and David Van Essen as the Senior Editor. The following individual involved in review of your submission agreed to reveal their identity: Hans Op de Beeck.

The reviewers have discussed the reviews with one another and the Reviewing Editor has drafted this decision to help you prepare a revised submission. Note that some of the essential revisions will require additional experimentation and/or significant new analyses. You may thus decide to forego resubmission should these requirements prove onerous. Moreover, as new results will be generated that could impact the conclusions, it remains possible that your manuscript will be rejected upon revision, should you choose to resubmit.

Summary:

This manuscript describes two fMRI experiments assessing shape processing in the ventral and dorsal pathways. Using two forms of scrambling, the authors assess BOLD responses in voxels and ROIs as a function of image coherence, with positive slopes reflecting greater responses to more shape information (i.e., lower levels of scrambling). They then examine how these slopes change as a function of position along the anterior-posterior axis, and find a dissociation between ventral (increasing then plateauing slopes) vs. dorsal (increasing then decreasing slopes) streams. Moreover, shape information in both ventral and dorsal streams correlated with object recognition behavior. The authors conclude that both ventral and dorsal streams contribute to perception, despite differences in how they code for shape information.

There was consensus that this study was rigorous, with two experiments, two manipulations, thorough analyses, good data reporting, and clear writing. Moreover, the findings were robust and interesting, speaking to an issue of fundamental importance about which there has been relatively little work.

At the same time, the reviewers raised several important and consistent concerns that would need to be addressed before a decision about publication can be reached. Below these concerns are described via constructive suggestions about how the manuscript would need to be improved to fare well in the next round.

Essential revisions:

1) Many of the claims relate to what has been learned about the nature of object representations in the dorsal stream. However, the reviewers question whether the approach taken here is adequate to assess representational content. In particular, the use of univariate methods and image scrambling to assess shape information was considered indirect and outdated, compared to the multivariate (or even adaptation) methods used in prior studies. Admittedly, the difference between scrambling techniques across streams (i.e., reduced shape sensitivity in dorsal stream in Experiment 2) speaks to what is being coded, but this was not viewed as sufficient. In particular, this can speak to selectivity for shape and/or identity, but does not address what features are being represented. Although adaptation would not be possible to consider with the current design, I see no reason that a representational similarity analysis couldn't be performed. There were 2 and 4 (Experiments 1 and 2) repetitions of each stimulus, which would allow a cross-correlation of the raw pattern of BOLD activity over searchlights or ROIs for each stimulus with its own repetition(s) and with the other stimuli. Average diagonal vs. off-diagonal correlations would provide an index of shape coding, and this could be examined at different levels of scrambling. Moreover, how this index changes over the longitudinal axis could provide confirmation of the piecewise regression results, and also about differences between scrambling techniques across streams. Finally, second-order correlations of the correlation matrices would speak directly to the question of the similarity between ventral and dorsal stream representations. Other multivariate approaches could achieve the same goals, and the authors are experts in this class of techniques, but something along these lines is necessary.

2) Related to questions about representation, it was unclear which exact properties of the objects were affected by scrambling. For example, reduced sensitivity to diffeomorphic scrambling in the dorsal stream is interpreted as evidence of coding for the presence of a single shape rather than identity. However, this kind of scrambling affects many other features (e.g., curvature, texture, etc.), all of which could be potential explanations. The convex hull analysis is a good start, but the authors should provide a more comprehensive, perhaps computational analysis of the features present in the stimuli from both experiments at different levels of scrambling.

3) This paper builds on a long history of related studies. The reviewers felt that this history and its implications were not adequately presented. This includes the classic work by Grill-Spector and others – and whether the current results would (or would not) have been predicted from it – as well as the more recent work using multivariate methods (e.g., by Bracci and colleagues). Many of these papers are cited, but a more in-depth weighing of the prior literature, discussion of the new findings and why they are novel in this historical context, and consideration of theoretical implications (including for previous conclusions) is needed.

4) The conclusion about differences in the representational gradient of ventral and dorsal streams depends on how the streams are defined. For example, the authors seem to go as anterior as possible in the dorsal stream and obtain a decrease in shape sensitivity. Would a decrease not be observed in the ventral stream if its end was likewise extended further, say into the hippocampus or anterior temporal lobe? Alternatively, if the dorsal stream was defined more conservatively, wouldn't the gradient match the ventral stream? The root concern is that it is unclear whether the gradients are fundamental, organizing properties of these streams, or at least partially artifacts of how the analysis was conducted. The authors need to better justify their definition of the streams, and address this limitation.

5) Relatedly, the anterior-posterior axis is defined simply as Y coordinates. Given the brain's curvature, especially in the dorsal stream, these coordinates do not reflect neural distance. The authors should calculate alternative distance metrics, for example tracing along the surface of inflated cortex from calcarine to central sulci, re-performing their piecewise regressions, and either replacing the existing analyses or reporting one in the supplement.

6) Object recognition performance is almost perfectly correlated with scrambling. As a result, the neural correlations with object recognition almost perfectly mirror the baseline relationships to scrambling. Thus, it is unclear what these results add to the story. More problematically, it is unclear whether neural activity is driven by scrambling or recognition. These can be dissociated, as Grill-Spector did with training, but that hasn't been done in the present manuscript. At a minimum, this issue should be highlighted in the Discussion; better would be an analytical solution to dissociating these factors.

[Editors' note: further revisions were requested prior to acceptance, as described below.]

Thank you for resubmitting your work entitled "The large-scale organization of shape processing in the ventral and dorsal pathways" for further consideration at *eLife*. Your revised article has been favorably evaluated by David Van Essen (Senior Editor), Nick Turk-Browne (Reviewing Editor), and one of the original reviewers.

The manuscript has been improved considerably and we plan to accept it if you can address a couple of remaining issues:

1) In light of the block design and mixing of exemplars, the added multivariate analysis is non-standard. Please acknowledge this more thoroughly before presenting the results, including by explaining how this differs from standard RSA over exemplars (which will be more familiar to readers).

2) Please also clarify your interpretation of the multivariate findings. Insofar as these regions represent exemplars and scrambling gradually destroys exemplar identity, why does collapsing over exemplars work at all? What aspects of the exemplar-general representations at each level of scrambling are similar? There are potentially uninteresting explanations, such as that scrambling changes the spatial extent of the image (especially for box scrambling). If you cannot rule out such accounts or provide a more parsimonious explanation, you might consider re-calibrating how much emphasis these findings receive and/or moving them to supplement. At a minimum, this analysis needs more motivation and interpretation.

3) It would be more informative to use Box Scrambling and Diffeomorphic Scrambling as subtitles rather than Experiment 1 and Experiment 2.

---

## [Author Response]

Essential revisions:1) Many of the claims relate to what has been learned about the nature of object representations in the dorsal stream. However, the reviewers question whether the approach taken here is adequate to assess representational content. In particular, the use of univariate methods and image scrambling to assess shape information was considered indirect and outdated, compared to the multivariate (or even adaptation) methods used in prior studies. Admittedly, the difference between scrambling techniques across streams (i.e., reduced shape sensitivity in dorsal stream in Experiment 2) speaks to what is being coded, but this was not viewed as sufficient. In particular, this can speak to selectivity for shape and/or identity, but does not address what features are being represented. Although adaptation would not be possible to consider with the current design, I see no reason that a representational similarity analysis couldn't be performed. There were 2 and 4 (Experiments 1 and 2) repetitions of each stimulus, which would allow a cross-correlation of the raw pattern of BOLD activity over searchlights or ROIs for each stimulus with its own repetition(s) and with the other stimuli. Average diagonal vs. off-diagonal correlations would provide an index of shape coding, and this could be examined at different levels of scrambling. Moreover, how this index changes over the longitudinal axis could provide confirmation of the piecewise regression results, and also about differences between scrambling techniques across streams. Finally, second-order correlations of the correlation matrices would speak directly to the question of the similarity between ventral and dorsal stream representations. Other multivariate approaches could achieve the same goals, and the authors are experts in this class of techniques, but something along these lines is necessary.

We thank the reviewers for this helpful comment. Note that, because we used a block design to increase the statistical robustness of our experiment, we cannot employ a multivariate analysis of the responses to the different exemplars. Nevertheless, following this comment, we have conducted a multivariate analysis (see below) and utilized RSA to uncover the sensitivity of the different ROIs to object scrambling and to compare the representational structure of the different ROIs.

The first-level RSA correlated the representational structure of the different ROIs with that of a shape-sensitivity model. Consistent with the results of the univariate analysis, we were able to demonstrate that the two- components organization along the longitudinal axis is not a unique property of the fMRI amplitude, but also emerges from the similarity of response patterns in the different ROIs.

Even more importantly, the second-level RSA allowed us to compare directly between the representational structures of the different ROIs. We used multidimensional scaling (MDS) plots (Figure 4 and Figure 5) to visualize the similarity and dissimilarity between the different ROIs and we statistically quantified the similarity of the representations within- and between-pathways. Interestingly, this analysis revealed that regions in the lateral ventral pathway derived similar representations to those derived by the posterior dorsal pathway, and that this, between-pathways similarity was even greater than the similarity observed between some regions in the same pathway. Hence, this analysis and the robust results further suggests that the neural representations of shape information are present in the two visual pathways and that these representations are broadly distributed in cortex.

The new analysis has been included in the revised manuscript (subsection “ii. Representational Similarity Analysis”) and Figure 4 and 5.

2) Related to questions about representation, it was unclear which exact properties of the objects were affected by scrambling. For example, reduced sensitivity to diffeomorphic scrambling in the dorsal stream is interpreted as evidence of coding for the presence of a single shape rather than identity. However, this kind of scrambling affects many other features (e.g., curvature, texture, etc.), all of which could be potential explanations. The convex hull analysis is a good start, but the authors should provide a more comprehensive, perhaps computational analysis of the features present in the stimuli from both experiments at different levels of scrambling.

We have conducted three additional image analyses to further quantify the differences between the diffeomorphic and the scrambling manipulations. These analyses enabled us to examine how the different distortion manipulations modulated low, mid and high-level shape attributes. Low-level attributes were examined using a pixel-similarity analysis. Mid-level attributes were mapped using image entropy (the amount of the disorder in an image) and image homogeneity (texture). High-level attributes were assessed using the Convex Hull algorithms. The results of these analyses revealed that object distortion was greater in box scrambling experiment compared with diffeomorphic experiment across all levels of shape information.

The continuity of object and image texture are essential to the existence of a “shape” and therefore it is hard to conclude whether the relative reduction of dorsal sensitivity to the diffeomorphic manipulation reflects sensitivity to “shape” per-se, or to shape elements such as continuous texture and a clear figure-ground segregation. We address this limitation in the Discussion section of the revised manuscript (subsection “What aspects of object properties are encoded by the dorsal pathway?”, fourth paragraph and see subsection “Comparison between stimuli as function of distortion method” for the analysis of shape attributes)

3) This paper builds on a long history of related studies. The reviewers felt that this history and its implications were not adequately presented. This includes the classic work by Grill-Spector and others – and whether the current results would (or would not) have been predicted from it – as well as the more recent work using multivariate methods (e.g., by Bracci and colleagues). Many of these papers are cited, but a more in-depth weighing of the prior literature, discussion of the new findings and why they are novel in this historical context, and consideration of theoretical implications (including for previous conclusions) is needed.

We have now considered previous investigations and the interpretation of the present findings in light of these previous findings. These discussions have been included in different sections of the revised manuscript. Thanks for drawing our attention to this inadvertent neglect of related work.

4) The conclusion about differences in the representational gradient of ventral and dorsal streams depends on how the streams are defined. For example, the authors seem to go as anterior as possible in the dorsal stream and obtain a decrease in shape sensitivity. Would a decrease not be observed in the ventral stream if its end was likewise extended further, say into the hippocampus or anterior temporal lobe? Alternatively, if the dorsal stream was defined more conservatively, wouldn't the gradient match the ventral stream? The root concern is that it is unclear whether the gradients are fundamental, organizing properties of these streams, or at least partially artifacts of how the analysis was conducted. The authors need to better justify their definition of the streams, and address this limitation.

Following this comment, we redefined the ventral pathway mask to include the anterior parts of the temporal lobe. This new definition extended the previous results observed for the ventral pathway. In particular, we have found that the ventral pathway, similar to the dorsal pathway, has a first positive component and a negative second component. Hence, the results obtained from the new, extended mask, may suggest that the topographical organization of the two pathways is shaped by similar factors, namely cortical distance from the visual cortex and connectivity to other cortical systems (i.e., the motor system in the dorsal pathway and semantic system in the ventral pathway).

Notably, even though the gradients are qualitatively similar (i.e. have 2 components, one positive and one negative), there is also evidence for differences in shape processing in the two pathways. First, in both experiments, the maximal shape sensitivity (the inflection point of the regression line) emerged in more posterior regions of the dorsal pathway compared with ventral pathway (and this effect holds even when we have used a different distance measurement that was computed based on both the Y and Z axes (see Author response image 1)). In addition, the average shape sensitivity is greater in the ventral pathway reflecting the centrality of this pathway in shape processing.

The results of the new analyses have been included in the revised manuscript (subsection “i. Univariate analysis”) and on Figure 2 and 3.

5) Relatedly, the anterior-posterior axis is defined simply as Y coordinates. Given the brain's curvature, especially in the dorsal stream, these coordinates do not reflect neural distance. The authors should calculate alternative distance metrics, for example tracing along the surface of inflated cortex from calcarine to central sulci, re-performing their piecewise regressions, and either replacing the existing analyses or reporting one in the supplement.

It is true that we focus our analysis on the Y (posterior-anterior) axis. This focus was theoretically driven and followed a previous related investigation (e.g., Mruczek et al., 2013; Freud et al., 2016). However, in light of the reviewers’ concerns, we have reanalyzed the data, taking into account both the Z (superior-inferior) and Y axes. To this end, we have calculated the distance between each voxels to the most posterior voxel (using both Y and Z coordinate). This approach replicates the main organization revealed using only the Y-axis (Author response image 1 and Figure 2—figure supplement 1 and Figure 3—figure supplement 1 of the revised manuscript). We still report the Y axis analysis for the sake of simplicity, but the distance analysis has been added and figures are included in the revised manuscript.

These results were reproduced when we used Principal Component Analysis to extract the common space of the two axes. The first PCA component (which explains ~89% of the variance) was correlated with the slope values and revealed similar organization to this described above.

**Author response image 1. respfig1:** Group-averaged piecewise regression analysis for Experiment 1 and 2. Each dot signifies a voxel, at a particular relative location based on the z and y coordinates, averaged across participants, and the black line shows the result of the piecewise regression (based on two components) obtained for the group average.

6) Object recognition performance is almost perfectly correlated with scrambling. As a result, the neural correlations with object recognition almost perfectly mirror the baseline relationships to scrambling. Thus, it is unclear what these results add to the story. More problematically, it is unclear whether neural activity is driven by scrambling or recognition. These can be dissociated, as Grill-Spector did with training, but that hasn't been done in the present manuscript. At a minimum, this issue should be highlighted in the Discussion; better would be an analytical solution to dissociating these factors.

We thank the reviewers for this important comment. To address this concern, we have calculated the partial correlations between behavioral performance and fMRI activation controlling for the correlation with the level of scrambling. Importantly, the partial correlation treats the level of scrambling as a covariate, and therefore allows us to reveal the unique correlation between fMRI activation and behavioral performance. The partial correlations were weaker (compared with the original, simple correlation) but were nevertheless, still evident and significant in the posterior parietal ROIs and in regions along the ventral pathway, pointing to a plausible contribution of dorsal regions to perception. The revised analysis is included in the subsection “iii. Correlation between fMRI activation and object recognition performance” and Figure 6.

[Editors' note: further revisions were requested prior to acceptance, as described below.]

The manuscript has been improved considerably and we plan to accept it if you can address a couple of remaining issues:1) In light of the block design and mixing of exemplars, the added multivariate analysis is non-standard. Please acknowledge this more thoroughly before presenting the results, including by explaining how this differs from standard RSA over exemplars (which will be more familiar to readers).

We have addressed this issue in the revised manuscript: “Notably, most previous investigations utilize the RSA approach to uncover the representational space of individual exemplars. […] This procedure still allows us to explore how the availability of shape information modulated the representational content of different ROIs.”

2) Please also clarify your interpretation of the multivariate findings. Insofar as these regions represent exemplars and scrambling gradually destroys exemplar identity, why does collapsing over exemplars work at all? What aspects of the exemplar-general representations at each level of scrambling are similar? There are potentially uninteresting explanations, such as that scrambling changes the spatial extent of the image (especially for box scrambling). If you cannot rule out such accounts or provide a more parsimonious explanation, you might consider re-calibrating how much emphasis these findings receive and/or moving them to supplement. At a minimum, this analysis needs more motivation and interpretation.

We agree with the reviewers that the interpretation of the RSA results needs to be thoughtfully considered, given the non-conventional approach we have adopted. In particular, since we have utilized a block design, we cannot reach any conclusions about the representational structure of any particular exemplar. Therefore, the RSA focuses on the extent to which the *availability of shape information* modulates the representational content in different ROIs (see comment above, subsection “ii. Representational Similarity Analysis”, first two paragraphs). You will recall that the experiment is blocked by level of scrambling: intact through 256 scrambled pieces. The RSA allows us to explore the extent to which each ROI represents information as a function of the coherence of the shape, and it is this parameter that enables us to reach a conclusion regarding the sensitivity to shape.

Another concern raised by the reviewers is that the similarities or dissimilarities between the representational structures of different ROIs might be accounted for by changes of low-level visual attributes, which are being distorted by the scrambling manipulation. It is for this very reason that we conducted the second experiment with the diffeomorphic manipulation and we conducted the RSA on these data as described above for the box scrambling manipulation i.e. we have applied the same analyses on fundamentally different scrambling manipulations (box scrambling and diffeomorphic scrambling) that distort object shape, but preserve different attributes of the original stimuli. Importantly, similar patterns of results for the first-order and second-order RSA were observed across the two experiments, reinforcing the notion that our results reflect the sensitivity of the visual system to shape information, rather than to low level attributes. This conclusion is also supported by the differences observed between the two experiments. In particular, the diffeomorphic scrambling preserves more shape information than the box scrambling manipulation and accordingly, the RSA for this experiment uncovered overall lower correlations with the shape model.

We have incorporated this discussion in the subsection “ii. Representational Similarity Analysis” and in the last paragraph of the subsection “What aspects of object properties are encoded by the dorsal pathway?”

3) It would be more informative to use Box Scrambling and Diffeomorphic Scrambling as subtitles rather than Experiment 1 and Experiment 2.

The paper has been revised accordingly.